# Dynamics-Predictive Sampling for Active RL Finetuning of Large Reasoning Models

**Yixiu Mao, Yun Qu, Qi Wang,**[*] **Heming Zou, Xiangyang Ji**[*]
Department of Automation, Tsinghua University
{myx21, qy22, zouhm24}@mails.tsinghua.edu.cn
cheemswang@mail.tsinghua.edu.cn, xyji@tsinghua.edu.cn

## Abstract

Reinforcement learning (RL) finetuning has become a key technique for enhancing the reasoning abilities of large language models (LLMs). However, its effectiveness critically depends on the selection of training data. Recent advances underscore the importance of online prompt selection methods, which typically concentrate training on partially solved or moderately challenging examples under the current policy, thereby yielding more effective model updates. While significantly accelerating RL finetuning in terms of training steps, they also incur substantial computational overhead by requiring extensive LLM rollouts over large candidate batches to identify informative samples, an expense that can outweigh the finetuning process itself. To address this challenge, this work proposes Dynamics-Predictive Sampling (DPS), which online predicts and selects informative prompts by inferring their learning dynamics prior to costly rollouts. Specifically, we introduce a new perspective by modeling each prompt's solving progress during RL finetuning as a dynamical system, where the extent of solving is represented as the state and the transition is characterized by a hidden Markov model. Using historical rollout reward signals, we perform online Bayesian inference to estimate evolving state distributions, and the inference outcome provides a predictive prior for efficient prompt selection without rollout-intensive filtering. Empirical results across diverse reasoning tasks, including mathematics, planning, and visual geometry, demonstrate that DPS substantially reduces redundant rollouts, accelerates the training process, and achieves superior reasoning performance. Our code is available at https://github.com/maoyixiu/DPS.

## 1 Introduction

Reinforcement learning (RL) finetuning has emerged as a crucial technique to enhance the reasoning capabilities of large language models (LLMs) (Lightman et al., 2023; Jaech et al., 2024; Guo et al., 2025; Team et al., 2025). These finetuned models, often referred to as large reasoning models (LRMs), generate chain-of-thoughts (CoTs) to perform multi-step structured inference and have achieved remarkable progress across a wide range of knowledge-intensive applications, including scientific question answering (He et al., 2024), symbolic mathematics (Luo et al., 2025b), logical deduction (Xie et al., 2025), and program synthesis (Luo et al., 2025a).

While RL finetuning has demonstrated substantial progress, its effectiveness depends heavily on the quality of training data (Guo et al., 2025; Yang et al., 2024b), prompting increasing attention to data curation (Wen et al., 2025; Hu et al., 2025). A common practice is to perform offline data filtering, in which prompts are ranked or selected prior to training using static heuristics such as estimated difficulty, domain balance, or diversity (Ye et al., 2025; Li et al., 2025; Wang et al., 2025b). Although beneficial, this approach fails to adapt to the model's evolving competence during training. To improve adaptivity, recent work has explored online prompt selection strategies that dynamically adjust to the model's evolving behavior. These methods typically operate on a per-step or per-epoch basis, selecting informative prompts that provide stronger training signals (Yu et al., 2025; Zhang

---

[*]Corresponding authors.

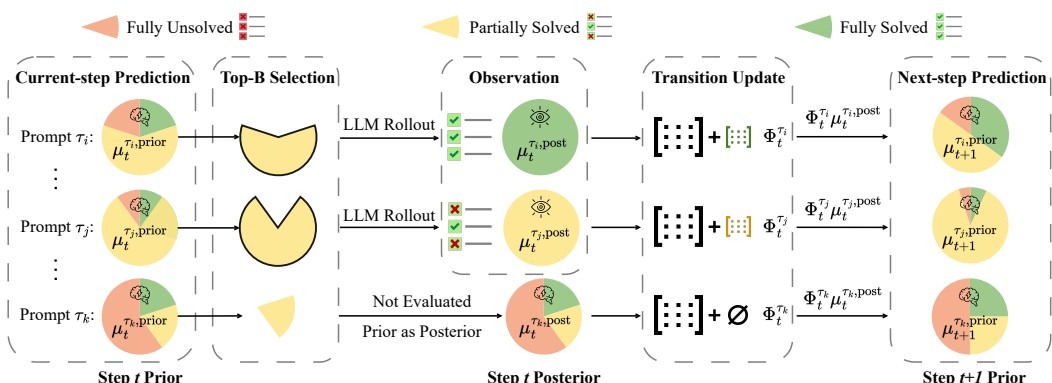

Figure 1: Dynamics-Predictive Sampling (DPS) framework. DPS models each prompt's solving progress in RL finetuning as a dynamical system, treating solving extent as the state with transitions characterized by a hidden Markov model. By employing lightweight inference, it predicts and selects informative (partially solved) prompts online, without requiring rollout-intensive filtering.

et al., 2025; Cui et al., 2025). A representative state-of-the-art (SoTA) approach is Dynamic Sampling (DS) (Yu et al., 2025), which expands candidate prompt batches, generates multiple responses per prompt, discards uninformative prompts with consistent rewards, and uses the retained subset for finetuning. This strategy improves training sample quality and significantly accelerates RL finetuning in terms of training steps. However, for reasoning-intensive tasks, generating responses with long CoTs is computationally expensive. As a result, DS incurs substantial overhead from extensive LLM generation on enlarged batches, which in practice often outweighs the cost of finetuning itself.

This work aims to preserve the adaptivity of online prompt selection while avoiding redundant rollouts. To this end, we propose Dynamics-Predictive Sampling (DPS), which online predicts informative prompts by inferring their learning dynamics. Specifically, we introduce a new perspective by formalizing each prompt's solving progress during RL finetuning as a dynamical system. The solving extent of each prompt is treated as the state of the system, while the distribution of these states evolves as LRM updates. Technically, this process is instantiated as a hidden Markov model (HMM), which serves as a tractable tool for tracking the prompt-solving dynamics. Given the constructed dynamical system, we perform online Bayesian inference to estimate the evolving state distributions from historical rollout reward signals. The inference outcome offers a predictive prior for adaptive prompt selection, thereby improving sample efficiency without rollout-intensive filtering.

Empirically, we evaluate the proposed DPS across diverse reasoning downstream tasks, including competition-level mathematics, numerical planning, and visual geometry. The results demonstrate that DPS can accurately predict prompts' evolving solving states and consistently select a higher proportion of informative samples compared to baseline methods. Leveraging this capability, DPS substantially accelerates RL finetuning, achieving performance comparable or even superior to the oracle rollout-intensive strategy DS with significantly fewer rollouts.

## 2 PRELIMINARY

**RL Finetuning for LRMs.** Given a prompt $\tau$ sampled from a dataset $\mathcal{D}$ and a response $y$ generated from the model's policy $\pi_{\boldsymbol{\theta}}(y|\tau)$, the objective of RL finetuning is to maximize the expected return:

$$\max_{\boldsymbol{\theta} \in \boldsymbol{\Theta}} \ \mathbb{E}_{\tau \sim \mathcal{D}, \ y \sim \pi_{\boldsymbol{\theta}}(\cdot|\tau)} \left[ r(\tau, y) \right], \tag{1}$$

where the reward function $r(\tau, y)$ typically verifies the correctness of responses, with binary signals commonly used in domains such as mathematics (i.e., 1 for correct and 0 for incorrect).

**Group Relative Policy Optimization (GRPO).** To solve the above optimization problem, a number of policy gradient methods have been proposed. GRPO (Shao et al., 2024) is a recent and widely adopted variant that eliminates the need for explicit value function estimation, making it particularly suitable for finetuning LLMs. Formally, for an arbitrary prompt $\tau$ and its corresponding $k$ sampled

responses $\{y_i^\tau\}_{i=1}^k$, GRPO maximizes the following objective:

$$\mathcal{J}_{\text{GRPO}}(\boldsymbol{\theta}) = \mathbb{E}_{\tau \sim \mathcal{D}, \{y_i^\tau\}_{i=1}^k \sim \pi_{\boldsymbol{\theta}_{\text{old}}}(\cdot|\tau)}$$

$$\left[ \frac{1}{k} \sum_{i=1}^k \left( \min \left( \frac{\pi_{\boldsymbol{\theta}}(y_i^\tau|\tau)}{\pi_{\boldsymbol{\theta}_{\text{old}}}(y_i^\tau|\tau)} \hat{A}_i^\tau, \text{ clip} \left( \frac{\pi_{\boldsymbol{\theta}}(y_i^\tau|\tau)}{\pi_{\boldsymbol{\theta}_{\text{old}}}(y_i^\tau|\tau)}, 1-\epsilon, 1+\epsilon \right) \hat{A}_i^\tau \right) - \beta D_{KL}(\pi_{\boldsymbol{\theta}}||\pi_{\text{ref}}) \right) \right],$$

where the clipped policy ratio prevents $\pi_{\boldsymbol{\theta}}$ from deviating excessively from the previous policy $\pi_{\boldsymbol{\theta}_{\text{old}}}$, while the regularization coefficient $\beta$ penalizes divergence from a fixed reference model $\pi_{\text{ref}}$. GRPO employs a group-based normalization scheme to estimate the advantages $\hat{A}_i^\tau$:

$$\hat{A}_i^\tau = \frac{r(\tau, y_i^\tau) - \text{mean}(\{r(\tau, y_j^\tau)\}_{j=1}^k)}{\text{std}(\{r(\tau, y_j^\tau)\}_{j=1}^k)}. \tag{2}$$

This strategy significantly reduces training complexity and has demonstrated strong empirical performance across diverse LLM reasoning tasks (Shao et al., 2024; Guo et al., 2025).

**Dynamic Sampling for Online Prompt Selection.** In RL finetuning of LLMs, training examples contribute unequally to policy improvement. When the model consistently answers a problem either correctly or incorrectly, a phenomenon frequently observed during training (Zhang et al., 2025), the reward provides limited optimization signals (Chen et al., 2025; Yu et al., 2025). For algorithms such as GRPO, this situation causes the normalized advantages to vanish, effectively halting optimization.

To mitigate this issue, online prompt selection strategies are proposed to dynamically curate prompts under specific rules (Zhang et al., 2025; Yu et al., 2025). A representative SoTA method is Dynamic Sampling (DS) (Yu et al., 2025). At each training step $t$, DS rolls out with a larger, randomly sampled candidate prompt batch $\hat{\mathcal{B}}_t$, and discards uninformative prompts with identical rewards across the $k$ responses, forming the final training batch $\mathcal{B}_t$:

$$\mathcal{B}_t = \left\{ \tau \in \hat{\mathcal{B}}_t \mid \text{std} \left( \{r(\tau, y_i^\tau)\}_{i=1}^k \right) > 0 \right\}. \tag{3}$$

Despite its effectiveness, DS introduces significant computational overhead due to repeated LLM rollouts and evaluations over the enlarged candidate batch. In many cases, the candidate batch is several times larger than the final batch, resulting in a proportional increase in LLM generation costs. This burden is particularly pronounced in reasoning tasks requiring long CoT generation.

For extended discussions on related work, we refer the reader to Appendix A.

## 3 Dynamics-Predictive Sampling for Active RL Finetuning

This section formalizes the prompt-solving progress as a dynamical system, develops an inference strategy for solving extent prediction, and proposes an efficient pipeline for online prompt selection.

### 3.1 Generative Modeling of Prompt-Solving Dynamics

**Problem Formulation.** Prior research has revealed the existence of prompt-solving states for efficient policy optimization. Specifically, History Resampling (HR) (Zhang et al., 2025) categorizes prompts into fully solved ones and others, whereas DS (Yu et al., 2025) distinguishes partially solved prompts from the rest. Both theoretical analyses and empirical findings (Bae et al., 2025; Chen et al., 2025) suggest that prompts yielding both successful and failed responses are more informative, as they provide stronger gradient signals for updates. In light of this, this work defines an implicit state $z_t^\tau \in \{1, 2, 3\}$ for each prompt $\tau \in \mathcal{D}$, indicating its rollout outcome at training step $t$:

- State 1 (fully unsolved): All responses are incorrect, $\sum_{i=1}^k r(\tau, y_i) = 0$;
- State 2 (partially solved): Some responses are correct and some incorrect, $0 < \sum_{i=1}^k r(\tau, y_i) < k$;
- State 3 (fully solved): All responses are correct, $\sum_{i=1}^k r(\tau, y_i) = k$.

According to prior work (Bae et al., 2025; Chen et al., 2025), State 2 prompts are the most informative and therefore should be prioritized during training. However, at each training step, the solving

state of any given prompt is unknown prior to rollout and evaluation. In the batch training setting, solving states are only observed intermittently, when certain prompts are selected for rollout. Consequently, each prompt yields an intermittent observation sequence, with the observation of prompt $\tau$ at step $t$ denoted as $y_t^\tau$ (where $y_t^\tau = \varnothing$ if no observation made). Our objective is to estimate the filtered prior belief of the solving state at step $t$ before observation, denoted by $\mu_t^{\tau,\text{prior}}$:

$$\mu_t^{\tau,\text{prior}}(i) := \mathbb{P}(z_t^\tau = i \mid y_{1:t-1}^\tau), \quad \forall i \in \{1,2,3\}. \tag{4}$$

**Prompt Solving as Dynamical Systems.** We formalize the evolution of each prompt's solving state using a Hidden Markov Model (HMM), which captures how the LLM's ability to solve a given prompt evolves during training. For clarity, we omit the superscript $\tau$ in this section and Section 3.2, describing the generative and inference process for a single prompt, which applies to all others.

Formally, the initial solving state $z_1$ is drawn from a categorical prior $\mu_1^{\text{prior}} \in \Delta^3$. In the absence of prior knowledge, we adopt a uniform distribution:

$$z_1 \sim \text{Categorical}(\mu_1^{\text{prior}}), \quad \mu_1^{\text{prior}} = \left[\tfrac{1}{3}, \tfrac{1}{3}, \tfrac{1}{3}\right]. \tag{5}$$

Subsequent states evolve according to a Markov process with a column-stochastic transition matrix $\Phi \in \mathbb{R}^{3\times3}$, where entry $\Phi(i,j)$ represents the probability of transitioning from state $j$ to state $i$:

$$z_t \mid z_{t-1} \sim \text{Categorical}(\Phi(\cdot, z_{t-1})), \quad \Phi(i,j) = \mathbb{P}(z_t = i \mid z_{t-1} = j), \quad \sum_{i=1}^{3} \Phi(i,j) = 1. \tag{6}$$

At each timestep, if the prompt is selected for training, the observation $y_t$ reveals the current state exactly; otherwise, the state remains unobserved. This yields a degenerate emission model:

$$p(y_t \mid z_t) = \begin{cases} \delta(y_t, z_t), & \text{if } y_t \in \{1,2,3\}, \\ 1, & \text{if } y_t = \varnothing, \end{cases} \tag{7}$$

where $\delta(\cdot, \cdot)$ denotes the Kronecker delta function. Assigning emission probability 1 to missing observations preserves marginal consistency while imposing no constraint on $z_t$. Putting these components together, the solving progress for each prompt can be represented as a dynamical system. Specifically, the joint distribution over states $z_{1:T}$ and observations $y_{1:T}$ factorizes as:

$$p(z_{1:T}, y_{1:T}) = \int p(z_1) \prod_{t=2}^{T} p(z_t \mid z_{t-1}, \Phi) \prod_{t=1}^{T} p(y_t \mid z_t) \mathrm{d}\Phi, \tag{8}$$

where the transition matrix $\Phi$ is treated as a random variable. This formulation specifies the underlying generative process, thereby enabling subsequent Bayesian inference over the solving states.

## 3.2 ONLINE INFERENCE AND TRANSITION LEARNING

We perform online Bayesian inference to track the solving states for a given prompt during training. The procedure follows a three-stage pipeline at each training step $t$: (i) update the prior $\mu_t^{\text{prior}}$ to a posterior $\mu_t^{\text{post}}$, using the observation $y_t$ if available, otherwise setting the posterior to the prior; (ii) if $y_t$ is observed, refine the transition model; and (iii) propagate the posterior forward through the transition model to generate the next-step prior $\mu_{t+1}^{\text{prior}}$.

**Observation Update.** If $y_t$ is observed, Bayes' rule updates the prior $\mu_t^{\text{prior}}$ to the posterior $\mu_t^{\text{post}}$:

$$\mu_t^{\text{post}}(i) = \frac{p(y_t \mid z_t = i)\, \mu_t^{\text{prior}}(i)}{\sum_k p(y_t \mid z_t = k)\, \mu_t^{\text{prior}}(k)} = \frac{\delta(y_t, i) \cdot \mu_t^{\text{prior}}(i)}{\sum_k \delta(y_t, k) \cdot \mu_t^{\text{prior}}(k)}, \quad \text{if } y_t \in \{1,2,3\}. \tag{9}$$

If $y_t$ is unobserved, the Bayesian update defaults to $\mu_t^{\text{post}} = \mu_t^{\text{prior}}$ without new evidence.

**Transition Update.** We place independent Dirichlet priors on the columns of transition matrix:

$$\Phi_t(\cdot, j) \sim \text{Dirichlet}(\alpha_t(1,j), \alpha_t(2,j), \alpha_t(3,j)), \quad \forall j \in \{1,2,3\}, \tag{10}$$

where $\alpha_t(i,j)$ specify the distribution over the transition probabilities. We initialize the transition matrix with an uninformative prior by setting $\alpha_0(i,j) = 1$. As observations arrive sequentially, the

parameters $\alpha_t(i,j)$ are updated online. Specifically, when $y_t$ is observed at step $t$, a Bayesian update is applied to $\alpha_t(i,j)$ using the soft transition statistics:

$$\alpha_t(i,j) = \alpha_{t-1}(i,j) + \xi_t(i,j), \tag{11}$$

where $\xi_t(i,j)$ denotes the posterior transition pseudo-count:

$$\xi_t(i,j) := \mathbb{P}(z_{t-1} = j, z_t = i \mid y_{1:t}), \quad \text{if } y_t \in \{1,2,3\}. \tag{12}$$

This update rule follows from the conjugacy between the Dirichlet and Categorical distributions. Observing a transition from state $j$ to $i$ adds one pseudo-count to the corresponding parameters of the Dirichlet prior. As the transition is uncertain, the expected contribution is given by $\xi_t(i,j)$. By the Markov property and the conditional independence of observations given states, we obtain:

$$\mathbb{P}(z_{t-1} = j, z_t = i \mid y_{1:t}) = \frac{\mu_{t-1}^{\text{post}}(j) \cdot \Phi_{t-1}(i,j) \cdot p(y_t \mid z_t = i)}{\sum_{j'} \mu_{t-1}^{\text{post}}(j') \sum_{i'} \Phi_{t-1}(i',j') \cdot p(y_t \mid z_t = i')}. \tag{13}$$

with derivations deferred to Appendix C. Using the deterministic emission model in Eq. (7), and setting $\xi_t = 0$ when $y_t$ is unobserved (so the Bayesian update defaults to the prior), $\xi_t$ simplifies to:

$$\xi_t(i,j) = \begin{cases} \dfrac{\mu_{t-1}^{\text{post}}(j) \cdot \Phi_{t-1}(i,j)}{\sum_{j'} \mu_{t-1}^{\text{post}}(j') \cdot \Phi_{t-1}(i,j')}, & \text{if } i = y_t, \\ 0, & \text{otherwise.} \end{cases} \tag{14}$$

*Non-stationary Extension.* The standard Bayesian HMM assumes stationary transition dynamics. However, prompt-solving states in LRMs may evolve non-stationarily due to the complex learning process. To accommodate changing transition dynamics, we propose a lightweight extension that applies an exponentially decayed Dirichlet posterior update to the transition model:

$$\alpha_t(i,j) = \lambda \cdot \alpha_{t-1}(i,j) + (1-\lambda) \cdot \alpha_0(i,j) + \xi_t(i,j), \quad \lambda \in (0,1). \tag{15}$$

This mechanism introduces forgetting by emphasizing recent transition statistics while gradually discounting outdated patterns. Smaller values of $\lambda$ yield faster adaptation to evolving dynamics. The prior $\alpha_0$ serves as a regularizer: it prevents collapse when recent evidence is sparse and also enables the encoding of domain knowledge about plausible transition structures.

**Next-state Prediction.** After the observation and transition updates at step $t$, we use the posterior belief $\mu_t^{\text{post}}$ and the inferred transition matrix $\Phi_t$ to form the predictive prior for the next step:

$$\mu_{t+1}^{\text{prior}} = \Phi_t \, \mu_t^{\text{post}}, \quad \text{i.e.,} \quad \mu_{t+1}^{\text{prior}}(i) = \sum_{j=1}^3 \Phi_t(i,j) \cdot \mu_t^{\text{post}}(j). \tag{16}$$

This prior $\mu_{t+1}^{\text{prior}}$ represents our forecast of the prompt-solving state at training step $t+1$ before its observation, and serves as the initial belief for the subsequent inference iteration. Unlike classical HMM smoothing methods (e.g., Forward-Backward (Baum et al., 1972)), which require access to full trajectories, our approach updates both the state belief and transition posterior in an online manner. Moreover, the computational cost of this inference framework is typically negligible compared to response rollout or model finetuning, as it involves only very low-dimensional matrix operations.

## 3.3 PROMPT SAMPLING WITH PREDICTED DYNAMICS

The central goal of modeling prompt-solving dynamics is to online predict which prompts should be prioritized for training at each step, before conducting costly rollouts. Given the predictive solving-state belief $\mu_t^{\tau,\text{prior}} = \mathbb{P}(z_t \mid y_{1:t-1})$ for each prompt $\tau$, we prioritize prompts according to their predicted probability of being partially solved (State 2), denoted $\mu_t^{\tau,\text{prior}}(2)$. Crucially, we rely on the prior belief $\mu_t^{\tau,\text{prior}}$ rather than the posterior $\mu_t^{\tau,\text{post}}$, since selection must occur before outcomes at step $t$ are observed via rollouts. Formally, the $B$ prompts with the highest State 2 probabilities are selected to constitute the training batch at step $t$:

$$\mathcal{B}_t = \text{Top}_B \left( \left\{ \tau \in \mathcal{D} \mid \mu_t^{\tau,\text{prior}}(2) \right\} \right). \tag{17}$$

We note that while the Top-B selection strategy is purely exploitative, it exploits an objective (i.e., $\mu_t^{\tau,\text{prior}}$) that already incorporates a degree of exploration via the non-stationary decay mechanism.

---

**Algorithm 1:** Dynamics-Predictive Sampling (DPS) for Active RL Finetuning

---

**Input:** Prompt dataset $\mathcal{D}$; Dirichlet prior $\alpha_0$; Initial state belief $\mu_1^{\text{prior}}$; Batch size $B$; Decay ratio $\lambda$; Large language model $\pi_{\boldsymbol{\theta}}$; Total training steps $T$.

**Output:** Finetuned large reasoning model $\pi_{\boldsymbol{\theta}}$.

**for** $t = 1$ **to** $T$ **do**

    // Select most likely informative prompts for training

    Sample a batch of prompts $\mathcal{B}_t \leftarrow \text{Top}_B\left(\left\{\tau \in \mathcal{D} \mid \mu_t^{\tau,\text{prior}}(2)\right\}\right)$;

    **foreach** $\tau \in \mathcal{B}_t$ **do**

        Generate $k$ responses using $\pi_{\boldsymbol{\theta}}$ and evaluate to obtain $y_t^\tau \in \{1, 2, 3\}$;

    Update the LLM $\pi_{\boldsymbol{\theta}}$ using trajectories from $\mathcal{B}_t$ with RL algorithm;

    // Update solving-state beliefs and transition dynamics

    **foreach** $\tau \in \mathcal{D}$ **do**

        **if** $y_t^\tau$ *is observed (i.e., $\tau \in \mathcal{B}_t$)* **then**

            Compute posterior belief $\mu_t^{\tau,\text{post}}$ via Bayes' rule by Eq. (9);

            Compute posterior transition pseudo-count $\xi_t^\tau$ by Eq. (14);

            Update Dirichlet transition posterior: $\alpha_t^\tau = \lambda \cdot \alpha_{t-1}^\tau + (1 - \lambda) \cdot \alpha_0^\tau + \xi_t^\tau$;

        **else**

            Set posterior belief $\mu_t^{\tau,\text{post}}$ to the prior belief $\mu_t^{\tau,\text{prior}}$;

            Decay Dirichlet transition posterior: $\alpha_t^\tau = \lambda \cdot \alpha_{t-1}^\tau + (1 - \lambda) \cdot \alpha_0^\tau$;

        Generate prior belief $\mu_{t+1}^{\tau,\text{prior}}$ for the next step by Eq. (16);

---

**Overall Algorithm.** Integrating these components, we present the complete algorithm DPS in Algorithm 1, with a framework overview shown in Fig. 1. A detailed analysis of the time complexity of DPS and its implicit connection to curriculum learning is provided in Appendix B.

## 4 EXPERIMENTS

In this section, we conduct several experiments to examine the validity of DPS. Appendices D, E, and F provide implementation details, additional results, and data examples, respectively.

### 4.1 EXPERIMENTAL SETUP

**Tasks.** We evaluate DPS across three challenging reasoning domains, training separate models on their respective datasets: competition-level mathematics (MATH dataset (Hendrycks et al., 2021)), numerical planning (Countdown dataset (Pan et al., 2025)), and visual geometric reasoning (Geometry3k dataset (Lu et al., 2021; Hiyouga, 2025)). To further assess its generality, we test a range of large language and multi-modal models that vary in capacity and architecture. Models are finetuned with the GRPO algorithm within the verl framework (Sheng et al., 2024) and evaluated by average Pass@1 accuracy over 16 completions per prompt. Details of the training datasets, test benchmarks, and base models are reported in Appendix D, with illustrative data examples in Appendix F.

**Baselines.** We compare against three sampling strategies: (i) Uniform Sampling (US): the default strategy that randomly selects prompts without preference. (ii) Dynamic Sampling (DS): a compute-intensive oracle approach that oversamples and filters prompts using rollout feedback (Yu et al., 2025). Here, "oracle" refers to sampling a batch of all partially solved prompts, instead of achieving the best performance by training on sampled prompts. (iii) History Resampling (HR): an heuristic method that excludes prompts from the dataset if they yield all correct responses in the current epoch (Zhang et al., 2025), effectively treating the fully solved state as absorbing at the epoch level.

### 4.2 PREDICTION ACCURACY OF PROMPT-SOLVING STATES

A key component of DPS is online prediction of each prompt's solving state, which enables adaptive prioritization of partially solved examples during training. We evaluate the accuracy of this predic-

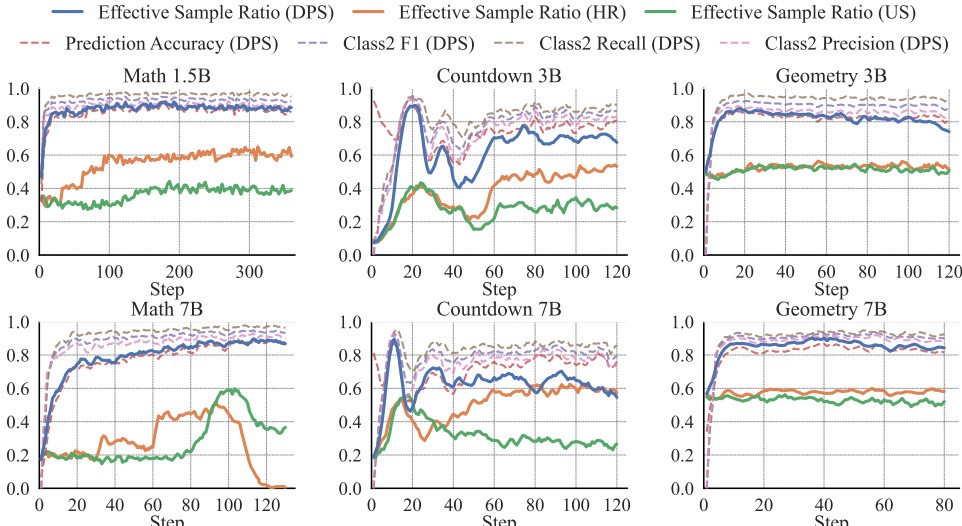

Figure 2: Proportion of partially solved prompts (Effective Sample Ratio) within sampled batches under different data sampling strategies, along with prediction metrics of DPS.

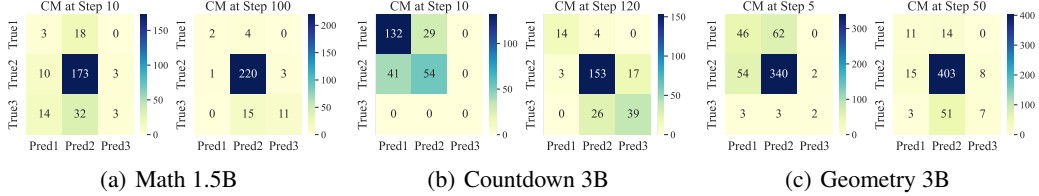

(a) Math 1.5B      (b) Countdown 3B      (c) Geometry 3B

Figure 3: Confusion Matrix (CM) for DPS predictions at different training steps across tasks.

tion mechanism by treating it as an online classification task. In Fig. 2, overall prediction accuracy is reported to assess general performance across the three classes, while precision, recall, and F1 score are additionally reported for Class 2 (partially solved), the state most critical for training efficiency. Throughout training, the predictor maintains high overall accuracy and achieves strong precision and recall for Class 2. Fig. 2 also shows the proportion of partially solved prompts in sampled batches. Compared with US and HR, DPS consistently yields a significantly higher concentration of such prompts, reaching approximately 90% in many tasks.

To further illustrate predictive behavior, Fig. 3 visualizes confusion matrices over training steps, where each cell gives the raw count for each (true, predicted) label pairs. Additional visualizations on more steps are deferred to Fig. 9. As training progresses, diagonal entries strengthen while off-diagonal errors diminish, showing improved discriminability. Notably, the center cell grows more prominent in both predictions and ground truth, indicating that the predictor increasingly emphasizes partially solved prompts. We also report the number of fully solved and unsolved prompts in batches across tasks in Fig. 8. Overall, these results demonstrate that DPS reliably tracks solving progress through lightweight inference and concentrates training on informative prompts.

## 4.3 RL FINETUNING EFFICIENCY AND PERFORMANCE

**Training Progress.** Fig. 4 presents the training curves of different sampling methods across tasks and models, where performance is tracked on AIME24 for MATH and on the respective test sets for Countdown and Geometry. DPS exhibits substantially faster policy improvement than US and HR and reaches higher final performance, benefiting from reliable prediction and a greater proportion of informative samples. In contrast, US and HR suffer degradation on MATH, likely due to entropy collapse (Liu et al., 2025a) arising from too few effective samples per batch. We attribute HR's less favorable performance to two factors: (i) its epoch-level absorbing transition assumption is overly rigid, limiting adaptability during training; and (ii) it only filters out fully solved prompts, which are often rare in early and middle stages of training. Relative to the oracle DS baseline, DPS achieves comparable overall performance across tasks and even slightly surpasses it on MATH. This advan-

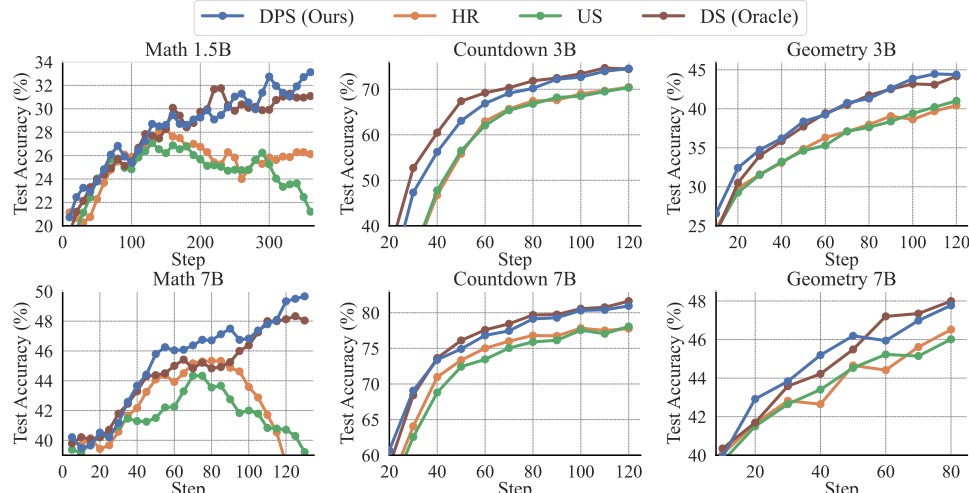

Figure 4: Training curves of different methods across reasoning tasks with varying model sizes. The curves in Math are smoothed with a window size of 5. DS serves as a high-resource oracle baseline.

Table 1: Evaluation across mathematics benchmarks. '+' represents finetuning with the method.

| Method | AIME24 | AMC23 | MATH500 | Minerva. | Olympiad. | Avg. ↑ | Rollouts↓ | Runtime↓ |
|---|---|---|---|---|---|---|---|---|
| R1-Distill-1.5B | 18.33 | 51.73 | 76.64 | 23.83 | 35.31 | 41.17 | - | - |
| +US | 26.46 | 63.18 | 82.78 | 27.46 | 43.00 | 48.57 | **737k** | **27h** |
| +HR | 28.13 | 64.61 | 82.88 | 27.37 | 43.15 | 49.23 | **737k** | 28h |
| +DS (Oracle) | 31.88 | 67.32 | 84.79 | **29.18** | **46.83** | 52.00 | 2933k | 89h |
| +DPS (Ours) | **32.71** | **67.77** | **84.95** | 29.09 | 46.11 | **52.13** | **737k** | 32h |
| R1-Distill-7B | 37.71 | 68.45 | 86.94 | 34.74 | 46.94 | 54.95 | - | - |
| +US | 45.83 | 73.57 | 89.06 | 37.68 | 50.42 | 59.31 | **287k** | **30h** |
| +HR | 46.46 | 75.98 | 90.01 | **37.94** | 51.50 | 60.38 | **287k** | 36h |
| +DS (Oracle) | 49.79 | 78.99 | 90.96 | 37.89 | 54.45 | 62.42 | 1147k | 73h |
| +DPS (Ours) | **51.04** | **80.35** | **91.13** | 37.82 | **55.32** | **63.13** | **287k** | 39h |

tage may stem from differences in sampling criteria: while DS samples randomly from evaluated partially solved prompts, DPS consistently selects the top-$B$ prompts with the highest predicted probability of being partially solved, which might be more beneficial for policy improvement.

**Generalization Performance.** We evaluate the trained models over multiple challenging benchmarks to assess their generalization capabilities. Table 1 reports results for models trained on MATH, evaluated on AIME24, AMC23, MATH500, MinervaMath, and OlympiadBench. The models are also evaluated on general reasoning benchmarks, including ARC-c and MMLU-Pro, with results provided in Table 4. Table 2 presents evaluations on Countdown, where models trained on a subset of the Countdown-34 dataset are tested on both the held-out split (CD-34) and a harder variant Countdown-4 (CD-4). Table 3 shows evaluations on Geometry. Across tasks, DPS consistently outperforms US and HR, while matching or exceeding DS in generalization performance.

Table 2: Evaluation on Countdown.

| Method | CD-34 | CD-4 | Rollouts |
|---|---|---|---|
| Qwen2.5-3B | - | - | - |
| +US | 69.87 | 39.42 | **246k** |
| +HR | 70.19 | 42.10 | **246k** |
| +DS (Oracle) | **74.95** | 47.67 | 1141k |
| +DPS (Ours) | 74.27 | **47.78** | **246k** |
| Qwen2.5-7B | - | - | - |
| +US | 77.84 | 53.27 | **246k** |
| +HR | 78.15 | 54.54 | **246k** |
| +DS (Oracle) | **81.26** | **60.77** | 1006k |
| +DPS (Ours) | 81.15 | 59.61 | **246k** |

**Rollout and Runtime Efficiency.** We also compare methods in terms of rollout usage and runtime. Tables 1, 2 and 3 report the total number of rollouts, while Fig. 11 plots the model performance as a function of rollout counts. The results demonstrate that DPS achieves strong performance with significantly fewer rollouts than DS, typically using less than 30% of DS's rollout budget to match or exceed its results. Moreover, as shown in Table 1, DPS incurs substantially lower runtime than DS when trained on the standard MATH dataset, generally using about half of DS's runtime. While

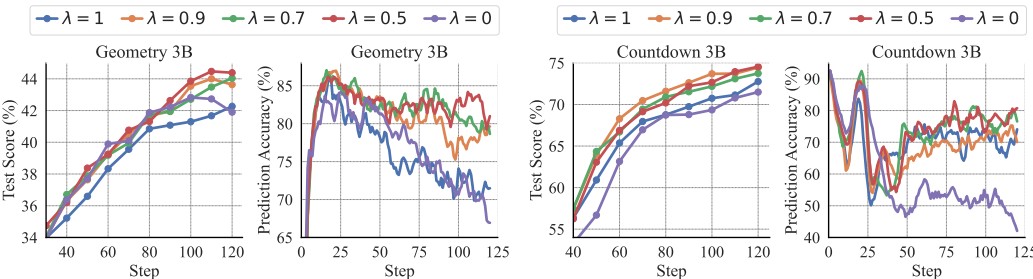

Figure 5: Performance and prediction accuracy of DPS under varying non-stationary decay ratios $\lambda$.

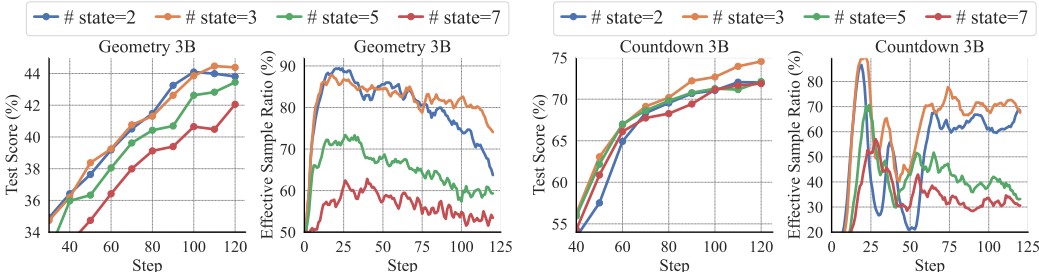

Figure 6: Performance and effective sample ratios of DPS under different solving-state partitions.

DPS exhibits slightly longer runtime than US and HR, this difference is not due to its prediction and selection operations, which are negligible in our experiments. Instead, it arises from longer response generations associated with higher performance, as illustrated in Fig. 13.

**Computational Scaling Behavior.** We further examine how the computational cost of different operations, including LLM training, LLM generation, and DPS sampling and updates, scales with both dataset size and LLM size. Detailed results and analyses are provided in Appendix E.6.

## 4.4 ABLATION STUDY

**Effects of Non-stationary Decay.** The non-stationary decay ratio $\lambda \in [0, 1]$ controls the extent to which older observations are gradually discounted. As shown in Fig. 5, DPS maintains strong performance over a wide range of $\lambda$ across tasks. Notably, removing non-stationary decay (i.e., $\lambda = 1$, which assigns equal weight to all past observations) results in a decline in both performance and prediction accuracy. This suggests that the solving-state dynamics is indeed non-stationary and that adaptation to recent observations is crucial. Conversely, setting $\lambda = 0$, which relies solely on the most recent feedback while discarding all past information, also leads to degraded performance and reduced prediction accuracy. A moderate decay ratio strikes a balance, allowing the model to remain responsive to recent trends while retaining sufficient historical context for robust estimation.

**Effects of Different Solving-State Partitions.** We examine the impact of coarser or finer partitions of solving states. With two states, prompts are divided into partially solved versus all others. With more than three states, the success rate interval $[0, 1]$ is uniformly partitioned, and prompts predicted to lie near $0.5$ accuracy are prioritized, as prior work (Bae et al., 2025) suggests these yield the most informative signals. Fig. 6 presents performance and effective sample ratios under different partitions, where the latter is still defined as the proportion of partially solved prompts in each batch. Overall, both metrics decline under either coarser or finer partitions. We attribute this to two factors: (i) coarse partitions that merge fully unsolved and fully solved prompts obscure their distinct dynamics, making transitions harder to model; and (ii) finer partitions distribute limited training observations across more states, resulting in sparsity and reduced prediction reliability.

**Effects of Transition Priors.** The transition prior $\alpha_0$ allows flexible incorporation of domain-specific knowledge about plausible transition patterns. The effects of different transition priors on prediction accuracy and training efficiency are analyzed in Appendix E.4.

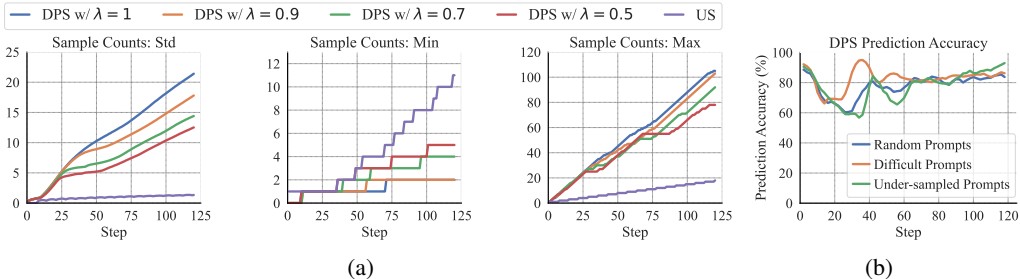

Figure 7: (a) Statistics of sample counts across the dataset on the Countdown 3B task. (b) DPS prediction accuracies for different types of prompts on the Countdown 3B task.

**Sensitivity to Response Group Size.** We evaluate DPS and US with response group sizes $k \in \{4, 8, 16\}$ on Countdown. Figs. 15 and 16 present learning curves of test accuracy, effective sample ratio, and DPS prediction accuracy. DPS consistently outperforms US in both performance and effective ratio, with the largest gap at $k = 4$. This is because a smaller $k$ reduces the probability that a policy produces a mix of correct and incorrect responses for a given prompt (with probability $1 - p^k - (1 - p)^k$ for success rate $p$), making US less likely to sample effective prompts, as reflected by its very low effective ratio at $k = 4$. DPS mitigates this by actively selecting effective prompts.

## 4.5 ADDITIONAL ANALYSIS

**Exploration in DPS.** Infrequently sampled prompts may have relatively inaccurate state predictions, potentially creating a negative feedback loop where they are sampled even less. However, the non-stationary decay in DPS (Eq. (15)) implicitly encourages exploration that mitigates this risk. As the transition posterior decays, predictions for under-sampled prompts drift toward a uniform distribution; when clearly informative prompts become scarce, these prompts are naturally revisited and updated. Fig. 7(a) supports this effect: a smaller decay parameter $\lambda$ yields more uniform sample counts, with lower variance and higher minimum across the dataset.

**Prediction Accuracy on Representative Prompts.** To examine prediction quality on representative prompts, after each training step, we evaluated three sets: (a) 256 prompts with the fewest past sample counts (under-sampled prompts), (b) 256 prompts with the highest DPS-estimated probability of being fully unsolved (difficult prompts), and (c) 256 randomly sampled prompts. Fig. 7(b) shows their prediction accuracies on the Countdown 3B task. Difficult prompts achieve even higher prediction accuracy than random ones, likely because they tend to exhibit simpler or more stable state distributions and transitions, making them easier to predict even with limited observations. Under-sampled prompts typically show slightly lower accuracy, but the gap is small. We conjecture that a similar explanation applies: under-sampled prompts are often confidently predicted to be in State 1 or 3, and thus may largely consist of very hard or very easy problems, which are generally easier to infer. In this sense, the prompts most susceptible to estimation error under infrequent sampling are often those whose states are inherently easier to predict. This provides an additional perspective on how DPS mitigates the impact of sparse observations.

## 5 CONCLUSION AND LIMITATIONS

This work models each prompt's solving progress during RL finetuning as a dynamical system, representing the solving extent as the state and characterizing its transition with a hidden Markov model. A lightweight inference strategy is developed to online predict and select informative prompts without rollout-intensive filtering. Empirical results across diverse reasoning tasks demonstrate that DPS reduces redundant rollouts, accelerates training, and achieves superior reasoning performance.

A limitation of this work lies in its reliance on correctness-based rewards to define solving states. Nevertheless, the DPS framework naturally extends to more complex reward structures, such as dense or process-based rewards, by partitioning cumulative return intervals. Furthermore, the use of the straightforward top-k selection strategy may not be optimal. Future work will explore more sophisticated criteria, such as entropy-based prioritization of uncertain samples.

## ACKNOWLEDGMENTS

This work was supported by the National Natural Science Foundation of China (NSFC) with the Number # 62306326 and the National Key R&D Program of China under Grant 2018AAA0102801. We thank all reviewers for their constructive feedback on this work.

## ETHICS STATEMENT

This work adheres to the ICLR Code of Ethics. All experiments use publicly available datasets, and no private, sensitive, or human-subject data are involved. The proposed methods focus on improving training efficiency and do not introduce additional ethical risks beyond standard LLM finetuning. We follow dataset licenses and ensure no privacy, safety, or fairness concerns arise.

## REPRODUCIBILITY STATEMENT

All theoretical derivations are provided in Appendix C. Full experimental details, including datasets, benchmarks, model configurations, evaluation metrics, RL finetuning procedures, sampling method implementations, and hyperparameters, are provided in Appendix D. All datasets used are public, and we are committed to releasing the complete code to support reproduction.

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

# A  RELATED WORK

**RL for LLM Optimization.**  Reinforcement learning (RL) has become a pivotal technique for adapting large language models (LLMs) to complex tasks and desired behaviors. In particular, Reinforcement Learning with Human Feedback (RLHF) has proven effective for aligning LLMs with human preferences and safety constraints (Ouyang et al., 2022; Dong et al., 2024; Rafailov et al., 2023; Dai et al., 2023; Sun et al., 2023; Sheng et al., 2024). In domains where reward signals are verifiable, such as mathematics, code generation, and symbolic planning, Reinforcement Learning with Verifiable Rewards (RLVR) has been shown to substantially enhance the reasoning capacity of LLMs (Jaech et al., 2024; Shao et al., 2024; Team et al., 2025; Chu et al., 2025; Guo et al., 2025). From an algorithmic perspective, Proximal Policy Optimization (PPO) (Schulman et al., 2017), a foundational policy gradient method in RL, is directly applicable to LLM finetuning. More recently, Group Relative Policy Optimization (GRPO) (Shao et al., 2024) eliminates the computational overhead of PPO's value network by introducing a lightweight, group-normalized advantage estimator, and has rapidly become one of the most widely used RL finetuning algorithms. Subsequent refinements have focused on mitigating gradient bias, reducing training instability, and lowering computational cost (Yuan et al., 2025; Yue et al., 2025; Liu et al., 2025b; Yu et al., 2025; Kazemnejad et al., 2024; Hu, 2025). On the application side, substantial efforts have extend RL finetuning to broader task domains and increasingly large-scale models (Luo et al., 2025b; Dang & Ngo, 2025; Luo et al., 2025a; Zeng et al., 2025; Meng et al., 2025; Xu et al., 2024). At the same time, infrastructure-level advances have developed scalable frameworks for distributed and compute-efficient RL training tailored to LLMs (Sheng et al., 2024; Hu et al., 2025).

**Data Selection for RL Finetuning.**  A growing body of work emphasizes that the effectiveness of RL finetuning critically depends on the quality of training data (Guo et al., 2025; Yang et al., 2024b), which has motivated growing interest in data curation as a driver of efficient learning (Hu et al., 2025; Wen et al., 2025; Wang et al., 2025a). A common approach is offline data filtering, which ranks or selects prompts prior to training based on static heuristics such as estimated difficulty, domain balance, or diversity (Ye et al., 2025; Li et al., 2025; Zhou et al., 2023; Wen et al., 2025; Hu et al., 2025; Yang et al., 2024b; Fatemi et al., 2025; Wang et al., 2025b). While beneficial, this approach introduces preprocessing overhead for ranking or clustering and, more importantly, fails to adapt to the model's evolving competence during training. To address this limitation, recent work has investigated online selection strategies that dynamically choose prompts in response to the model's current behavior (Yu et al., 2025; Zhang et al., 2025). One class of methods performs per-step selection, either by filtering out uninformative prompts (Yu et al., 2025; Liu et al., 2025a; Cui et al., 2025; Meng et al., 2025) or by focusing on examples of intermediate difficulty (Bae et al., 2025). While these strategies improve the quality of training samples, they remain hindered by the high computational cost of rollout-intensive filtering or by limited accuracy in difficulty estimation. Alternative approaches adopt per-epoch data selection, updating the sample set periodically (Zhang et al., 2025; Zheng et al., 2025). However, these methods typically rely on coarse heuristics or empirical trends observed over epochs, which limits their responsiveness and often introduces high estimation error. Concurrently with this work, some studies (Qu et al., 2025; Shen et al., 2025) are motivated by a similar predict-then-sample principle, in which intermediate-difficulty prompts are prioritized before rollout. These methods typically model each prompt's success rate as a latent variable and formulate prompt selection as a Bernoulli bandit problem. However, this approach is better suited to settings with relatively stable success rates and is less flexible than DPS in capturing complex dynamics underlying continual model evolution. Moreover, under sparse observations, it lacks a reliable mechanism for extrapolation over unobserved intervals. Our approach formalizes prompt-solving progress as a dynamical system and introduces a tractable inference strategy for step-wise prompt selection with negligible computational overhead, achieving accurate prediction, fast convergence, and superior performance under a low rollout budget.

# B  DISCUSSIONS

**Time Complexity.**  We analyze the time complexity of Uniform Sampling (US), DS (Yu et al., 2025), and DPS. DS repeatedly samples candidate prompts, performs LLM rollouts, and discards those that fail to meet predefined constraints until $|\mathcal{B}|$ prompts are retained. Let $p_{\text{keep}}$ denote the expected probability that a sampled prompt is retained in DS, $C_{\text{llm}}$ the expected cost for generating

and evaluating $k$ LLM rollouts per prompt, $C_{\text{pred}}$ the expected cost of inference per prompt in DPS, and $C_{\text{topk}}$ the expected cost of top-k selection over the dataset in DPS.

The expected time complexity for prompt selection and evaluation per step is: $\mathcal{O}\left(|\mathcal{B}|C_{\text{llm}}\right)$ for US, $\mathcal{O}\left(\lceil \frac{1}{p_{\text{keep}}} \rceil |\mathcal{B}|C_{\text{llm}}\right)$ for DS, and $\mathcal{O}\left(|\mathcal{D}|C_{\text{pred}} + C_{\text{topk}} + |\mathcal{B}|C_{\text{llm}}\right)$ for DPS. Since our method involves only very low-dimensional matrix operations ($C_{\text{pred}}, C_{\text{topk}} \ll C_{\text{llm}}$), it holds that $\mathcal{O}\left(|\mathcal{D}|C_{\text{pred}} + C_{\text{topk}} + |\mathcal{B}|C_{\text{llm}}\right) \approx \mathcal{O}\left(|\mathcal{B}|C_{\text{llm}}\right)$. Therefore, DPS significantly reduces computational overhead compared to DS while typically adding negligible cost relative to the default US. The prediction and selection overhead in DPS scales approximately linearly with the dataset size $|\mathcal{D}|$. For existing popular datasets, this overhead is negligible. However, for potential extremely large datasets where the cost may become non-trivial, one can approximate the full-dataset updates and selection using a randomly sampled candidate subset $\hat{\mathcal{B}}$ ($|\mathcal{B}| < |\hat{\mathcal{B}}| < |\mathcal{D}|$) at each step.

**Implicit Curriculum Learning.** Beyond maximizing learning signals, this selection strategy induces an implicit form of curriculum learning (Bengio et al., 2009). Early in training, prompts with high State 2 probability are typically easier ones, for which the model begins to show partial success. As training progresses and the model improves, these prompts may transition to the fully solved state (State 3) and are no longer selected. Conversely, harder prompts that were initially always incorrect (State 1) may begin to yield partially correct responses, making them eligible for sampling.

This mechanism creates a self-paced progression from easier to harder prompts: beginning with tractable examples to bootstrap learning, then gradually shifting to more challenging cases as model capacity grows. Moreover, by targeting prompts in the partially solved regime, the method avoids both trivial and unsolvable cases, which provide little training benefit and may waste resources. Crucially, this adaptive curriculum is not manually curated but emerges naturally from the method, providing a principled and scalable alternative to handcrafted curricula.

## C   PROOF AND DERIVATION

**Derivation of the Transition Update.** The posterior transition pseudo-count $\xi_t(i,j)$ is defined for observed emissions $y_t \in \{1, 2, 3\}$ as:

$$\xi_t(i,j) := \mathbb{P}(z_{t-1} = j, z_t = i \mid y_{1:t}), \quad \text{if } y_t \in \{1, 2, 3\}. \tag{18}$$

The joint posterior distribution can be expressed as:

$$\mathbb{P}(z_{t-1} = j, z_t = i \mid y_{1:t}) = \frac{\mathbb{P}(z_{t-1} = j, z_t = i, y_{1:t})}{\mathbb{P}(y_{1:t})} \tag{19}$$

Using the Markov property $z_t \perp y_{1:t-1} \mid z_{t-1}$ and the conditional independence of observations $y_t \perp y_{1:t-1}, z_{t-1} \mid z_t$, the numerator factorizes as:

$$\mathbb{P}(z_{t-1} = j, z_t = i, y_{1:t}) = \mathbb{P}(y_{1:t-1}, z_{t-1} = j) \cdot \mathbb{P}(z_t = i \mid z_{t-1} = j) \cdot \mathbb{P}(y_t \mid z_t = i). \tag{20}$$

Substituting into the posterior expression yields:

$$\mathbb{P}(z_{t-1} = j, z_t = i \mid y_{1:t}) = \frac{\mathbb{P}(y_{1:t-1}, z_{t-1} = j) \cdot \mathbb{P}(z_t = i \mid z_{t-1} = j) \cdot \mathbb{P}(y_t \mid z_t = i)}{\mathbb{P}(y_{1:t})} \tag{21}$$

$$= \frac{\mathbb{P}(z_{t-1} = j \mid y_{1:t-1}) \cdot \mathbb{P}(z_t = i \mid z_{t-1} = j) \cdot \mathbb{P}(y_t \mid z_t = i)}{\mathbb{P}(y_t \mid y_{1:t-1})} \tag{22}$$

$$\tag{23}$$

Using the notation $\mu_{t-1}^{\text{post}}(j) := \mathbb{P}(z_{t-1} = j \mid y_{1:t-1})$, $\Phi_{t-1}(i,j) := \mathbb{P}(z_t = i \mid z_{t-1} = j)$, and $p(y_t \mid z_t = i) := \mathbb{P}(y_t \mid z_t = i)$, we obtain:

$$\mathbb{P}(z_{t-1} = j, z_t = i \mid y_{1:t}) = \frac{\mu_{t-1}^{\text{post}}(j) \cdot \Phi_{t-1}(i,j) \cdot p(y_t \mid z_t = i)}{\mathbb{P}(y_t \mid y_{1:t-1})}. \tag{24}$$

The normalizing denominator $\mathbb{P}(y_t \mid y_{1:t-1})$ can be obtained by marginalization:

$$\mathbb{P}(y_t \mid y_{1:t-1}) = \sum_{j'} \mu_{t-1}^{\text{post}}(j') \sum_{i'} \Phi_{t-1}(i', j') \cdot p(y_t \mid z_t = i'). \tag{25}$$

Therefore, the full expression becomes:

$$\mathbb{P}(z_{t-1} = j, z_t = i \mid y_{1:t}) = \frac{\mu_{t-1}^{\text{post}}(j) \cdot \Phi_{t-1}(i, j) \cdot p(y_t \mid z_t = i)}{\sum_{j'} \mu_{t-1}^{\text{post}}(j') \sum_{i'} \Phi_{t-1}(i', j') \cdot p(y_t \mid z_t = i')}. \tag{26}$$

Under the deterministic emission model in Equation (7), we have $p(y_t \mid z_t = i) = \delta(y_t, i)$ for $y_t \in \{1, 2, 3\}$, where $\delta$ denotes the Kronecker delta function. Substituting gives:

$$\mathbb{P}(z_{t-1} = j, z_t = i \mid y_{1:t}) = \frac{\mu_{t-1}^{\text{post}}(j) \cdot \Phi_{t-1}(i, j) \cdot \delta(y_t, i)}{\sum_{j'} \mu_{t-1}^{\text{post}}(j') \sum_{i'} \Phi_{t-1}(i', j') \cdot \delta(y_t, i')}, \quad \text{if } y_t \in \{1, 2, 3\}. \tag{27}$$

Note that $\delta(y_t, i')$ is non-zero only when $i' = y_t$, so the inner sum over $i'$ reduces to $\Phi_{t-1}(y_t, j')$. Thus, the expression simplifies to:

$$\mathbb{P}(z_{t-1} = j, z_t = i \mid y_{1:t}) = \begin{cases} \dfrac{\mu_{t-1}^{\text{post}}(j) \cdot \Phi_{t-1}(i, j)}{\sum_{j'} \mu_{t-1}^{\text{post}}(j') \cdot \Phi_{t-1}(i, j')} & \text{if } i = y_t, \ y_t \in \{1, 2, 3\}, \\ 0 & \text{if } i \neq y_t, \ y_t \in \{1, 2, 3\}. \end{cases} \tag{28}$$

Setting $\xi_t = 0$ for unobserved $y_t$ so that the Bayesian update in Equations (11) and (15) defaults to the prior without new evidence, the expression of $\xi_t$ simplifies to:

$$\xi_t(i, j) = \begin{cases} \dfrac{\mu_{t-1}^{\text{post}}(j) \cdot \Phi_{t-1}(i, j)}{\sum_{j'} \mu_{t-1}^{\text{post}}(j') \cdot \Phi_{t-1}(i, j')}, & \text{if } i = y_t, \\ 0, & \text{otherwise}. \end{cases} \tag{29}$$

# D  EXPERIMENTAL DETAILS

## D.1  DETAILS OF TASKS AND MODELS

We evaluate DPS across three distinct and challenging reasoning domains: competition-level mathematics, numerical planning, and visual geometric reasoning. To verify its broad applicability, we experiment with a range of large language and multi-modal models with varying capacities and architectures. We adopt the popular GRPO algorithm implemented within the verl framework (Sheng et al., 2024) to fine-tune models. Evaluation is based on average `pass@1` accuracy computed over 16 independent completions per example. Training datasets, test benchmarks, and base models in each domain are detailed as follows, with illustrative data examples provided in Appendix F.

### D.1.1  MATHEMATICS

**Training Dataset.** For mathematics, we train large reasoning models on the training split of MATH dataset (Hendrycks et al., 2021), consisting of 7,500 problems designed to reflect competition-level difficulty. Specifically, we use the Hugging Face release from `https://huggingface.co/datasets/DigitalLearningGmbH/MATH-lighteval`, consistent with prior work (Sheng et al., 2024).

**Test Benchmarks.** We assess performance across diverse mathematics benchmarks including AIME24, AMC23, MATH500 (Lightman et al., 2023), Minerva Math (Lewkowycz et al., 2022), and OlympiadBench (He et al., 2024), with all the datasets obtained from DeepScaler (Luo et al., 2025b). In particular, AIME24 is used to monitor training progress and plot the training curves. We additionally evaluate the trained models on general reasoning benchmarks, including ARC-c (Clark et al., 2018) and MMLU-Pro (Wang et al., 2024). We follow the evaluation setup in Yan et al. (2025) and adopt PRIME's prompt template for evaluation.

**Base Models.** Following prior work (Luo et al., 2025b), two base models from DeepSeek (Guo et al., 2025) are used: `DeepSeek-R1-Distill-Qwen-1.5B` from Hugging Face repository `https://huggingface.co/deepseek-ai/DeepSeek-R1-Distill-Qwen-1.5B` and `DeepSeek-R1-Distill-Qwen-7B` from `https://huggingface.co/deepseek-ai/DeepSeek-R1-Distill-Qwen-7B`.

### D.1.2 NUMERICAL PLANNING

**Training Dataset.** For arithmetic planning, we use the Countdown Number Game, where agents must construct the target number using basic operations over a given number set (Pan et al., 2025). Training is carried out on a 2,000-item subset of the complete Countdown-34 dataset at Hugging Face repository `https://huggingface.co/datasets/Jiayi-Pan/Countdown-Tasks-3to4`.

**Test Benchmarks.** Models are evaluated on two benchmarks: (i) CD-34, containing 512 held-out problems from Countdown-34; (ii) CD-4, including 512 problems from Countdown-4, a harder generalization version that operates 4 numbers, accessible at `https://huggingface.co/datasets/Jiayi-Pan/Countdown-Tasks-4`. In particular, CD-34 is used to monitor training progress and plot the training curves.

**Base Models.** Following prior work Chen et al. (2025), we test with two base models from Qwen (Yang et al., 2024a): `Qwen2.5-3B` from `https://huggingface.co/Qwen/Qwen2.5-3B` and `Qwen2.5-7B` from `https://huggingface.co/Qwen/Qwen2.5-7B`.

### D.1.3 VISUAL GEOMETRY

**Training Dataset.** Visual geometry experiments leverage the training split of the Geometry3k dataset (Lu et al., 2021; Hiyouga, 2025), accessible from `https://huggingface.co/datasets/hiyouga/geometry3k`. The dataset comprises 2,101 diagram-based geometry questions, requiring both image understanding and symbolic reasoning.

**Test Benchmark.** We evaluate trained models on the benchmark test set comprising 601 visual reasoning problems.

**Base Models.** For visual geometric reasoning, we adopt two vision-language models from Qwen (Bai et al., 2025): `Qwen2.5-VL-3B-Instruct` from `https://huggingface.co/Qwen/Qwen2.5-VL-3B-Instruct` and `Qwen2.5-VL-7B-Instruct` from `https://huggingface.co/Qwen/Qwen2.5-VL-7B-Instruct`.

## D.2 IMPLEMENTATION DETAILS

**RL Finetuning Implementations.** Our method and all sampling baselines shared the same RL finetuning implementations, detailed as follows. We adopt the popular GRPO algorithm (Shao et al., 2024) implemented within the verl framework (Sheng et al., 2024) to fine-tune models. Evaluation is based on average `pass@1` accuracy computed over 16 independent completions per prompt sampled with temperature 0.6 and nucleus sampling parameter `top_p` $= 0.95$, following the setup of Luo et al. (2025b). For each training step, we generate $k = 8$ responses per prompt under temperature 1.0 and `top_p` $= 1.0$ to compute advantage estimates and finetune models. An entropy regularization term with weight 0.001 is introduced, consistent with Luo et al. (2025b). Models is optimized with Adam (Kingma & Ba, 2014), using a constant learning rate of $1e{-}6$, momentum parameters $(0.9, 0.999)$, no warm-up, and weight decay of 0.01. We further adopt the Clip-Higher scheme in DAPO (Yu et al., 2025), which employs asymmetric clipping bounds, $\epsilon_{\text{low}} = 0.2$ and $\epsilon_{\text{high}} = 0.28$.

Task-specific training configurations are as follows: batch size is set to 256 for MATH (mini-batch 128) and Countdown (mini-batch 64), and to 512 for Geometry3k (mini-batch 256). The maximum output length is set to 8192 tokens for MATH and 1024 tokens for Countdown and Geometry3k. The KL-divergence penalty is omitted in actor loss for MATH and Countdown, following (Yu et al., 2025), but preserved in Geometry3k to maintain stable optimization, with a coefficient of 0.01 for 3B models and 0.03 for 7B models. For MATH, we use a binary reward function that assigns a reward

of $1$ for a correct answer and $0$ otherwise, following the default setup in verl (Sheng et al., 2024), while for Countdown and Geometry3k, we include a format bonus of $0.1$ in the reward function if the response is incorrect but with correct formatting, following the setup in Pan et al. (2025).

All experiments are executed on 8 NVIDIA A100 GPUs with 80GB memory.

**Sampling Method Implementations.** For Dynamic Sampling (DS) (Yu et al., 2025), we directly use the implementation from verl (Sheng et al., 2024), where prompts with zero variance in rewards are filtered out at each training step. For History Resampling (HR) (Zhang et al., 2025), we implement it within the verl framework by excluding prompts from the training dataset if they yield all correct responses in the current epoch. For DPS, we initialize the state belief as $\mu_1^{\text{prior}}(i) = 1/3$ and set the Dirichlet prior as $\alpha_0(i, j) = 1$, assuming no prior knowledge about both the initial prompt-solving states and transition probabilities. Thus, the only hyperparameter that requires tuning is the non-stationary decay ratio $\lambda$, which is set to $0.7$ for MATH and $0.5$ for Countdown and Geometry3k.

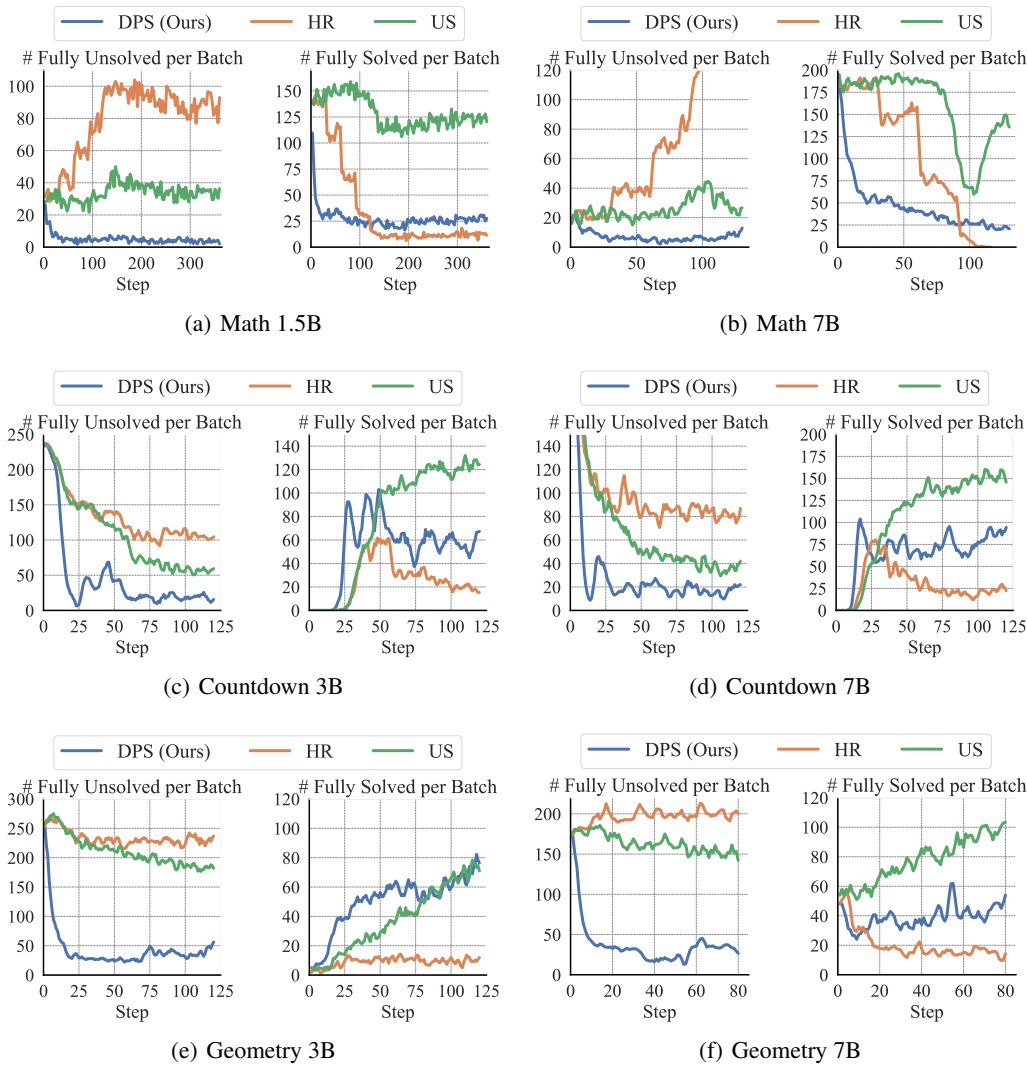

Figure 8: Number of ineffective prompts (fully unsolved or fully solved) in training batches sampled with different strategies across tasks.

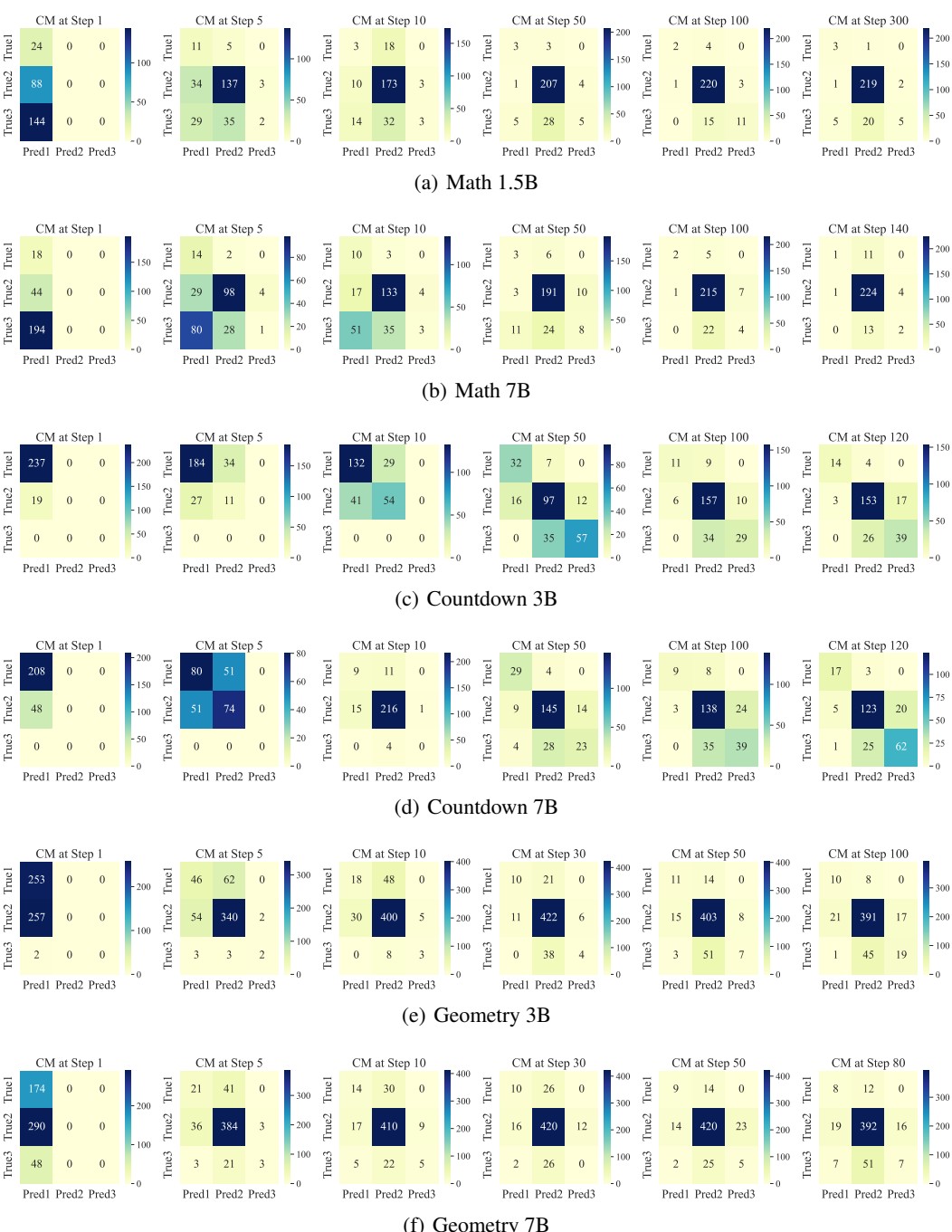

Figure 9: Confusion Matrix (CM) for DPS prediction at different training steps across tasks.

# E    EXTENDED EXPERIMENTAL RESULTS

## E.1    ADDITIONAL PREDICTION RESULTS

A key component of DPS is the real-time prediction of each prompt's solving state, which enables adaptive prioritization of partially solved examples during training. We evaluate the accuracy of this prediction mechanism by treating it as a dynamic classification task. This section provides additional analysis to complement Section 4.2.

**Number of Fully Solved and Unsolved Prompts.** Figure 8 reports the number of fully solved and fully unsolved prompts in batches across tasks. The results show that DPS consistently and significantly yields fewer fully solved and fully unsolved prompts than US across all tasks. In addition, HR treats the fully solved state as absorbing, which is much stricter than that of DPS. As a result, HR produces the fewest fully solved prompts across tasks but also the largest number of fully unsolved prompts. Overall, this leads to a substantially lower effective sample ratio for HR compared to DPS, as shown in Figure 2.

**Confusion Matrix.** Figure 9 visualizes confusion matrices over training steps across tasks, where each cell shows the raw count for each (true, predicted) label pair. As training progresses, diagonal entries strengthen while off-diagonal errors diminish, indicating improved discriminability. Notably, the center cell becomes increasingly prominent in both predictions and ground truth, suggesting that the predictor places greater emphasis on the target region.

Overall, these results demonstrate that DPS reliably tracks solving progress through lightweight inference and concentrates training on desired prompts.

### E.2 ADDITIONAL EVALUATION RESULTS

**Evaluation on Geometry.** Table 3 shows evaluations on the Geometry task, where models are trained and tested on the respective official Geometry3k datasets. The results show that DPS outperforms US and HR under the same rollout budget. On the other hand, DPS matches DS while requiring significantly fewer rollouts, making it more scalable in practical settings.

Table 3: Evaluation results on Geometry.

| Method | Qwen2.5-VL-3B-Instruct | | Qwen2.5-VL-7B-Instruct | |
| | Test Score ↑ | Rollouts ↓ | Test Score ↑ | Rollouts ↓ |
| --- | --- | --- | --- | --- |
| US | 40.69 | **492k** | 46.22 | **328k** |
| HR | 40.44 | **492k** | 46.52 | **328k** |
| DS (Oracle) | 44.33 | 1262k | **48.11** | 782k |
| DPS (Ours) | **44.47** | **492k** | 47.78 | **328k** |

**Evaluation on General Reasoning Benchmarks.** We additionally evaluate the MATH-trained models on general reasoning benchmarks, including ARC-c (Clark et al., 2018) and MMLU-Pro (Wang et al., 2024). We follow the evaluation setup in Yan et al. (2025) and adopt PRIME's prompt template for evaluation. The results are provided in Table 4. On these general (OOD) reasoning tasks, DPS also shows consistent improvements over the baseline methods.

Table 4: Evaluation on general reasoning benchmarks for models trained on the MATH dataset. Performance is measured by Pass@1 accuracy with a maximum response length of 8k tokens. '+' represents finetuning with the method.

| Method | ARC-c | MMLU-Pro | Avg. ↑ | Rollouts ↓ | Runtime ↓ |
| --- | --- | --- | --- | --- | --- |
| R1-Distill-1.5B | 41.81 | 21.02 | 31.42 | - | - |
| +US | 43.17 | 21.24 | 32.21 | **737k** | **27h** |
| +HR | 42.83 | 21.03 | 31.93 | **737k** | 28h |
| +DS (Oracle) | 44.88 | 23.25 | 34.07 | 2933k | 89h |
| +DPS (Ours) | **46.16** | **23.41** | **34.79** | **737k** | 32h |
| R1-Distill-7B | 74.32 | 50.44 | 62.38 | - | - |
| +US | 75.09 | 50.59 | 62.84 | **287k** | **30h** |
| +HR | 74.57 | 51.56 | 63.07 | **287k** | 36h |
| +DS (Oracle) | 77.05 | 51.43 | 64.24 | 1147k | 73h |
| +DPS (Ours) | **78.67** | **52.37** | **65.52** | **287k** | 39h |

**Evaluation with the Llama Model.**    Beyond Qwen-series models, we further train Llama-3.2-3B-Instruct on Countdown to evaluate different sampling methods. Figure 10 compares the resulting test accuracies and effective sample ratios. The results show that, with Llama-3.2-3B-Instruct, DPS also performs comparably to DS and surpasses HR and US in both test accuracy and effective sample ratios, with even larger relative gains than those observed with the Qwen models.

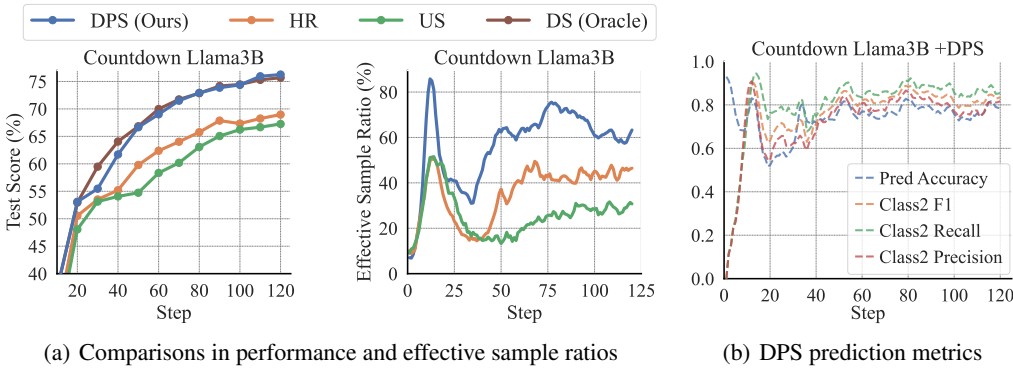

(a) Comparisons in performance and effective sample ratios    (b) DPS prediction metrics

Figure 10: Comparisons of different sampling methods using the additional model Llama-3.2-3B-Instruct. (a) Performance and effective sample ratios. (b) Prediction metrics of DPS.

**Evaluation with Extended Response Length.**    We further explore generalization by conducting an out-of-distribution study: MATH models, trained with a maximum response length of 8k, are tested under an extended 32k response budget. The results are reported in Table 5. DPS not only continues to surpass US and HR, but also slightly outperforms DS, showing clear advantages from the increased response length. These results highlight the scalability and generalization capacity of DPS in large-context settings.

Table 5: Evaluation across mathematics benchmarks under a maximum response length of 32k. '+' represents finetuning with the method. Evaluation is based on average Pass@1 accuracy over 16 responses per prompt.

| Method | AIME24 | AMC23 | MATH500 | Minerva. | Olympiad. | Avg. ↑ | Rollouts ↓ | Runtime ↓ |
|---|---|---|---|---|---|---|---|---|
| R1-Distill-1.5B | 28.12 | 61.67 | 83.18 | 26.54 | 43.33 | 48.57 | - | - |
| +US | 31.46 | 67.70 | 84.22 | 27.94 | 45.06 | 51.28 | **737k** | **27h** |
| +HR | 30.42 | 66.49 | 84.30 | 27.53 | 45.06 | 50.76 | **737k** | 28h |
| +DS (Oracle) | 32.92 | 69.95 | **86.44** | **30.26** | **49.66** | 53.85 | 2933k | 89h |
| +DPS (Ours) | **37.92** | **71.16** | 85.84 | 29.14 | 48.32 | **54.48** | **737k** | 32h |

### E.3    ROLLOUT EFFICIENCY

Figure 4 demonstrates that DPS and DS significantly accelerate RL finetuning over US and HR in terms of training steps. Yet, such comparisons overlook the cost of LLM rollout inference, which often exceeds finetuning itself. Because DS depends on oversampling, it rolls out a larger batch of prompts per training step, which substantially increases LLM inference overhead. Figure 11 plots performance against rollout numbers during training. The results show that DPS reaches strong performance with far fewer rollouts than DS, typically requiring less than 30% of DS's rollout budget to match or surpass its results.

### E.4    EFFECTS OF TRANSITION PRIORS

Our approach allows flexible incorporation of inductive bias by modifying the Dirichlet prior over the transition matrix. While the default configuration uses an uninformative prior $\alpha_0(i, j) = 1$ for all $(i, j)$, many real-world scenarios may exhibit structural regularities in their solving dynamics. This section investigates how certain priors affect prediction accuracy and training efficiency.

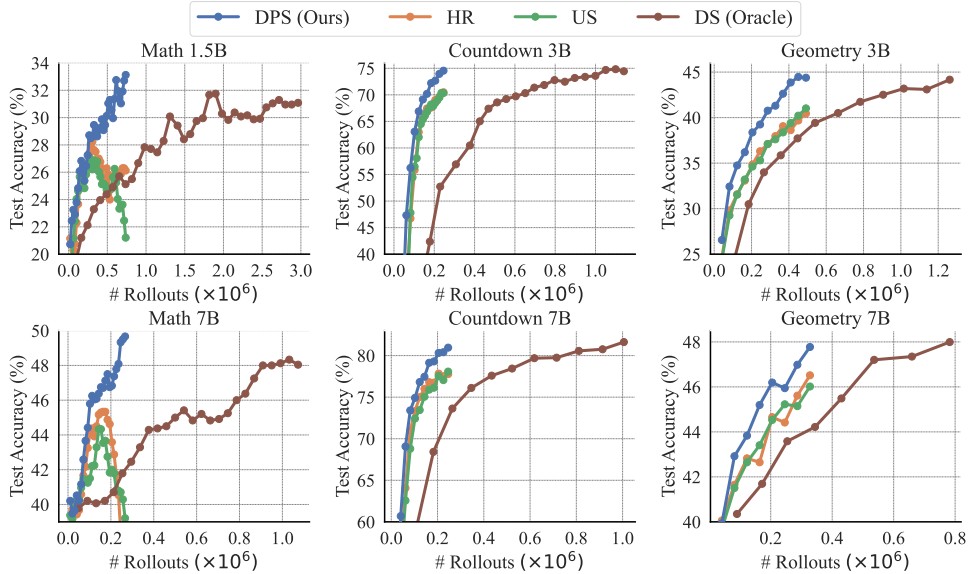

Figure 11: Training curves over the number of rollouts generated by LLM during training.

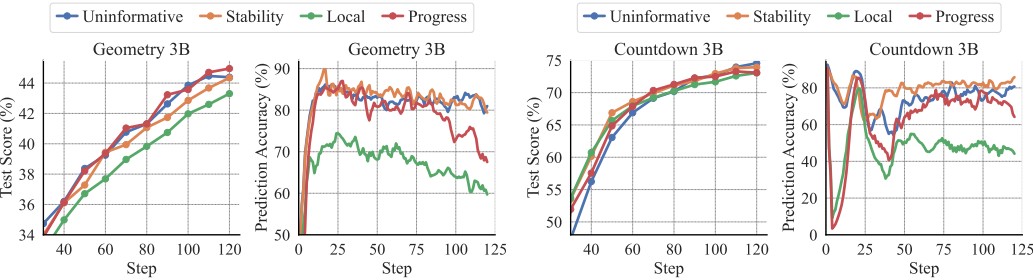

Figure 12: Performance and prediction accuracy of DPS under different transition priors.

We evaluate several representative priors, each encoding a different structural assumption: (i) Stability prior (stability-promoting): Assigns larger pseudo-counts to self-transitions ($\alpha_0(i,i) = 1$, $\alpha_0(i,j) = 0.5$ for $i \neq j$), which suppresses frequent state changes and reflects a belief that solving states tend to persist across steps. (ii) Progress prior (anti-regression): Sets lower pseudo-counts for regression transitions ($\alpha_0(i,j) = 0.5$ for $i < j$), imposing a preference against regressing from a more solved state to a less solved one. (iii) Local prior (local-transition): Sets $\alpha_0(i,j) = 0$ for $|i - j| > 1$, suppressing long-range transitions while retaining flexibility for adjacent-state updates. This encodes an assumption of smooth, gradual evolution in solving dynamics.

These priors are evaluated under identical training settings, with results shown in Figure 12. We report both task performance and prediction accuracy. We find that certain structured priors can lead to slight improvements over the uninformative baseline, particularly during early training stages where data is limited (see Countdown for example). As training progresses and more data becomes available, the advantages of structural priors diminish with degraded prediction accuracy, due to a potential mismatch between the prior's bias and the actual dynamics. This highlights the tradeoff between introducing prior structure and maintaining long-term flexibility.

## E.5 RESPONSE AND PROMPT LENGTH

**Response Length.** Response length has been identified as a strong correlate of reasoning ability (Yu et al., 2025). Figure 13 illustrates how different strategies influence this metric during MATH training. The average response length of DPS initially aligns with US and HR but quickly

increases, following a trajectory similar to DS. This also suggests that DPS rapidly learns the underlying prompt-solving dynamics. Both DPS and DS generate responses that are consistently longer than those from US. Longer outputs provide opportunities for deeper exploration and enable the model to engage in more complex reasoning processes, which may partly explain the observed performance gap (Yu et al., 2025). Notably, in the MATH 7B setting, HR exhibits a sharp increase in response length during later training stages. We attribute this to HR's rigid exclusion rule: once a prompt is fully solved at some epoch, it is permanently removed, even if errors may occur later. Under the stronger 7B model, this removes too many relatively easy problems, substantially raising the average difficulty of the remaining set. Faced with unsolvable inputs, the model tends to generate excessively long responses, often approaching the length limit (Hou et al., 2025).

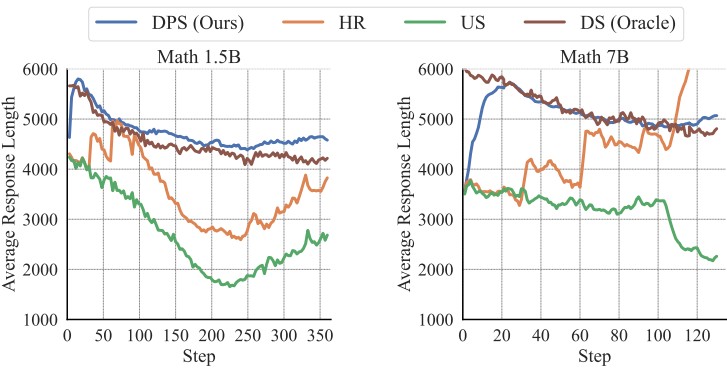

Figure 13: Average response length in the sampled batch during MATH training.

**Prompt Length.**    Figure 14 tracks the average length of sampled prompts throughout MATH training. Compared with US, all of DPS, HR, and DS tend to select longer prompts, and the average prompt length increases slightly as training progresses. This trend can be explained as follows. Since DS and DPS target partially solved prompts, improvements in trained model competence shift the training batches toward more difficult prompts, which are statistically often longer. Likewise, HR's exclusion of already fully solved examples leaves a progressively harder pool of prompts, also corresponding to greater length on average.

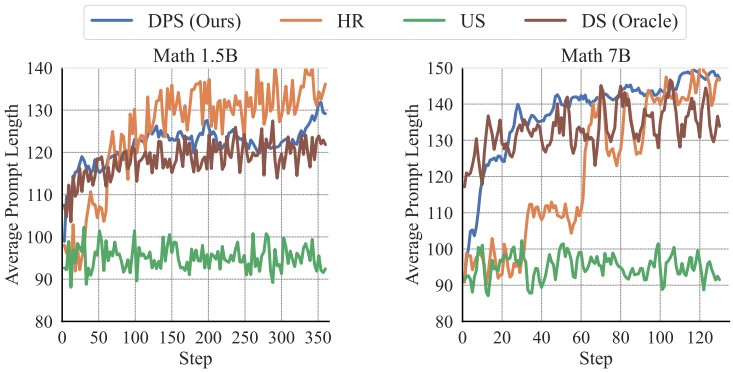

Figure 14: Average prompt length in the sampled batch during MATH training.

### E.6    EMPIRICAL ANALYSIS ON COMPUTATIONAL SCALING BEHAVIOR

This section conducts experiments to examine how the computational cost of different operations scales with both dataset size and LLM size.

(1) Computational scaling with dataset size. We construct pseudo-datasets (with arbitrary size $|\mathcal{D}|$) to more systematically evaluate the cost of DPS sampling and updates. Specifically, at each step

$t$, we randomly generate $|\mathcal{D}|$ transition posterior matrices $\alpha_t^\tau$ and belief vectors $\mu_t^{\tau,\text{post}}$ and $\mu_t^{\tau,\text{prior}}$ corresponding to all $|\mathcal{D}|$ pseudo-samples. In the sampling stage, we perform top-$B$ selection on $\mu_t^{\tau,\text{prior}}(2)$; in the HMM-update stage, we assign random observations to the batch of $B$ samples and apply independent HMM updates to all $|\mathcal{D}|$ samples. For comparison, the per-step costs of LLM training and generation, which are independent of dataset size, are obtained by finetuning the 7B model on MATH. Table 6 reports the per-step costs of different operations for dataset sizes ranging from $10^4$ to $10^7$. The runtime and memory usage of DPS scale approximately linearly with dataset size, yet even for a very large dataset of size $|\mathcal{D}| = 10^7$, DPS requires only 2.4s of runtime and 0.9 GiB of memory, while consuming no GPU memory. In contrast, LLM training and generation together require about 1100s of runtime and 600 GiB of GPU memory.

Given its linear scaling, the computational overhead of DPS could become non-negligible at a sufficiently large scale ($|\mathcal{D}| > 10^8$), though such dataset sizes are beyond typical practical settings. For these cases, Appendix B also discusses a scheme that approximates the full-dataset updates and selection using a randomly sampled candidate subset $\hat{\mathcal{B}}$ satisfying $B < |\hat{\mathcal{B}}| \ll |\mathcal{D}|$.

Table 6: Computational cost of different operations across varying dataset sizes, measured by per-step runtime and memory usage during the finetuning of DeepSeek-R1-Distill-Qwen-7B (8 A100 GPUs, batch size 256). The results for LLM training and generation are evaluated on the MATH dataset, while those for DPS are obtained on pseudo-datasets that emulate large-scale scenarios.

| | LLM train | LLM generation | DPS (sample + update) | | | |
|---|---|---|---|---|---|---|
| Dataset size | any | any | $10^4$ (MATH) | $10^5$ | $10^6$ | $10^7$ |
| Runtime (s) | 580 | 520 | 0.0005+0.002 | 0.004+0.02 | 0.06+0.2 | 0.6+1.8 |
| Memory (GiB) | $\approx$600 (GPU) | $\approx$600 (GPU) | $\approx$0.0009 | $\approx$0.009 | $\approx$0.09 | $\approx$0.9 |

(2) Computational scaling with LLM size. The cost of LLM training and generation scales with model size. In particular, the additional rollout cost of DS also grows with LLM size, whereas DPS, as a rollout-free alternative to DS, incurs no such dependence. Table 7 compares the per-step costs of different operations for 1.5B and 7B models. The total runtime of LLM training and generation increases from roughly 370s to 1100s as the model size increases from 1.5B to 7B. At the 7B scale, the additional overhead introduced by DS versus DPS is approximately 1500s vs. 0.003s. Therefore, the advantage of DPS can become increasingly significant as LLM size grows.

Table 7: Computational cost of different operations across varying LLM sizes, measured by per-step runtime for finetuning on the MATH dataset (8 A100 GPUs, batch size 256). The 1.5B and 7B models refer to DeepSeek-R1-Distill-Qwen-1.5B and DeepSeek-R1-Distill-Qwen-7B, respectively.

| | LLM train | | LLM generation | | DS sample (baseline) | | DPS (sample + update) |
|---|---|---|---|---|---|---|---|
| Model size | 1.5B | 7B | 1.5B | 7B | 1.5B | 7B | any |
| Runtime (s) | 170 | 580 | 200 | 520 | $\approx 3\times200$ | $\approx 3\times520$ | 0.0005+0.002 |

### E.7 SENSITIVITY ANALYSIS ON THE RESPONSE GROUP SIZE

We evaluate DPS and US under different response group sizes $k \in \{4, 8, 16\}$ on the Countdown 3B task. Figures 15 and 16 present the learning curves of test accuracy, effective sample ratio, and DPS prediction accuracy. The results show that DPS consistently outperforms US with both higher performance and effective sample ratios, and the advantage of DPS is the most pronounced when $k = 4$. For US, the performance with $k = 4$ drops substantially compared to $k = 8$ and 16 (falling to less than half), whereas DPS exhibits only a slight decrease (about $4\%$). In particular, for $k = 4$, the test accuracy of DPS is more than twice that of US.

We attribute this to the fact that for smaller $k$, the probability that the same policy produces a mix of correct and incorrect responses for the same prompt becomes lower (given a fixed success rate $p$, the probability of generating mixed responses is $1 - p^k - (1 - p)^k$). Hence, with a smaller $k$, the default US is much less likely to sample effective prompts (reflected in the extremely low effective sample ratio of US at $k = 4$ in Figure 16). This creates greater potential for improvement when using DPS, which actively selects effective prompts. On the other hand, a smaller $k$ may lead to

more frequent state transitions and make the underlying dynamics harder to estimate. Nevertheless, as shown in Figure 16, DPS maintains high prediction accuracy at the small-yet-practical value of $k = 4$, leading to substantial performance gains. Consequently, in scenarios where the response group size is constrained, such as under limited training resources, applying DPS is likely to be particularly advantageous.

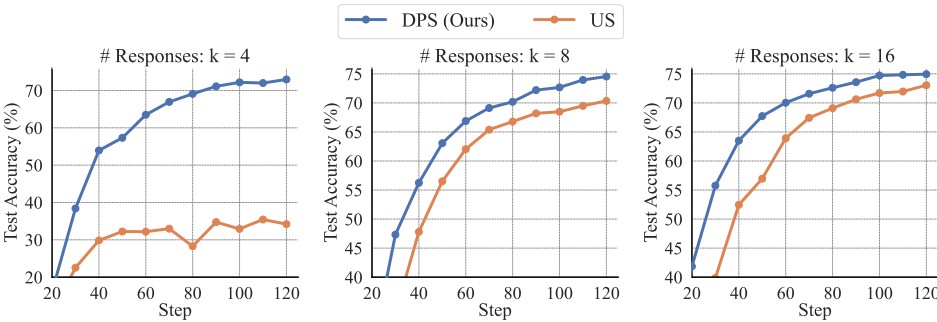

Figure 15: Performance of DPS and Uniform Sampling (US) under different response group sizes on the Countdown 3B task.

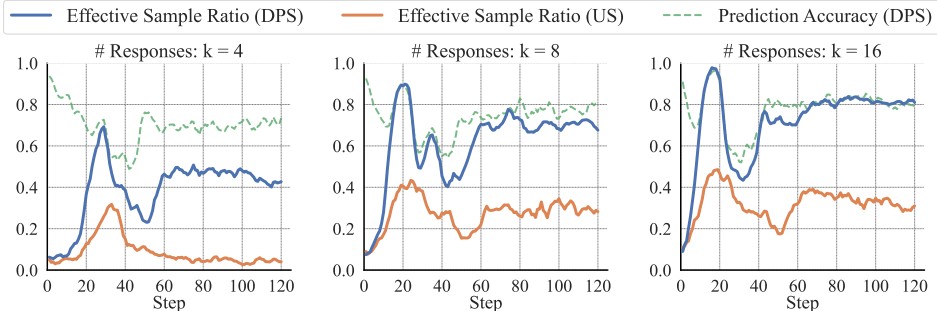

Figure 16: Effective sample ratios and prediction accuracies under different response group sizes on the Countdown 3B task.

### E.8 ENTROPY REGULARIZED SELECTION SCHEME

Introducing exploration into sample selection could potentially improve the model's robustness. To this end, we test a variant, DPS+Entropy, that explicitly balances exploitation and exploration by combining the entropy of the predicted distribution with the State-2 probability for Top-B sampling. We conduct experiments on Countdown and tune the entropy regularization coefficient in $\{0.01, 0.1, 1, 10\}$. The training curves are shown in Figure 17. DPS+Entropy performs best when the coefficient is $0.1$, but it does not yield a noticeably greater improvement over DPS in either test accuracy or effective sample ratio. We provide further analysis below.

While the Top-B selection strategy is purely exploitative, it exploits an objective (i.e., the predicted probability) that already incorporates a degree of exploration. The non-stationary decay mechanism in DPS (Eq. (15)), although originally designed to accommodate non-stationary dynamics, also implicitly introduces exploration. It gradually decays the transition posterior and drifts the predicted states of under-sampled prompts (i.e., those predicted to be in State 1 or 3) toward a more uniform distribution, increasing their likelihood of being selected and updated. This behavior is supported by Figure 7(a), which shows that a smaller decay ratio $\lambda$ leads to more uniform sample counts, with a lower variance and a higher minimum across the dataset. Hence, the additional entropy term partly overlaps with this built-in exploration effect, which may account for the limited improvement.

We also note that the specific choice of selection criterion is not the primary focus of this work. Once the state distribution is predicted, any selection criterion, such as softmax selection or entropy-based

sampling, can be applied. DPS adopts a simple criterion and introduces as few hyperparameters as possible (with only $\lambda$) while already achieving strong performance.

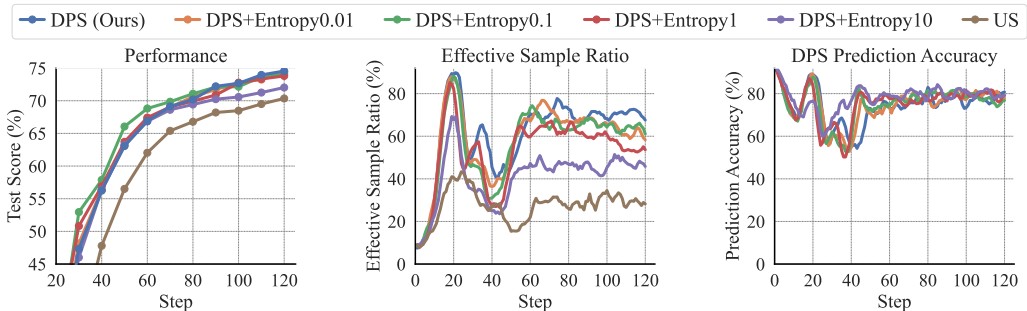

Figure 17: Evaluation of the entropy regularized selection scheme for DPS on Countdown 3B task.

### E.9 COMPARISON WITH ADDITIONAL BASELINES

**Simple Non-probabilistic Heuristic.** We implemented a simple predictive baseline, denoted Var+EMA, that tracks an exponential moving average of the reward variance for each prompt and samples the prompts with the Top-B values across the dataset. We conduct experiments on Countdown with Var+EMA, tuning the EMA smoothing factor in $\{0, 0.1, 0.5, 0.9\}$ and choosing $0.5$ as it yields relatively better performance. The comparative results in Figure 18 show that DPS outperforms Var+EMA with higher test accuracy and effective sample ratios. The following analyzes the necessity and advantages of the HMM framework over this simple predictive heuristic. (i) Dynamics estimation. Var+EMA implicitly assumes that the solving extent of each prompt tends to persist across steps, which resembles maintaining a fixed, stability-promoting transition model in DPS. Therefore, this heuristic is less flexible than DPS in capturing more complex underlying dynamics that may arise in practice. (ii) State prediction. Due to the infrequent sampling of a given prompt, its reward-variance observations are unavailable on most steps. Under this setting, Var+EMA lacks a reliable mechanism to extrapolate and predict variance during these unobserved intervals. In contrast, a core advantage of the HMM framework is its ability to model state transitions and, crucially, to extrapolate under missing observations. Regarding hyperparameters, DPS uses only one parameter, the non-stationary decay ratio $\lambda$, whereas Var+EMA uses an EMA smoothing factor.

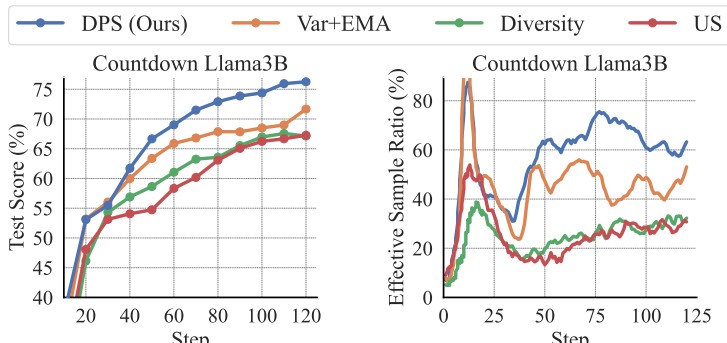

Figure 18: Comparison with additional baselines in terms of performance and effective sample ratio.

**Diversity-based Sampling.** We also implement a baseline that performs active sampling based on batch-level sample diversity. Specifically, we first pre-sample a candidate batch that is $n$ times larger than the actual training batch, and embed each prompt into a 1024-dimensional vector using WordLlama. We then iteratively select the candidate whose embedding maximizes the cumulative pairwise $L_2$ distance to previously selected samples, thereby greedily constructing a batch with high dispersion in the embedding space. We evaluate this variant on Countdown and tune the candidate

batch size multiplier $n \in \{2, 4, 8\}$, ultimately selecting $n = 4$ as it yields slightly better performance. As shown in Figure 18, diversity-based sampling offers only marginal improvements over US, and both its test accuracy and effective sample ratio remain far below those of DPS.

### E.10 PRELIMINARY EXPLORATION OF EXTENSIONS TO CONTINUOUS PROCESS REWARDS

This section first discusses the main challenges of applying active sampling in process-reward settings, and then presents a preliminary exploration of extending DPS to continuous process rewards.

Our focus on binary rewards reflects their practical prevalence and their well-understood connection with sample informativeness, which enables principled sampling strategies. In contrast, how process rewards relate to informativeness remains unclear in the field. To our knowledge, existing methods that incorporate process rewards still rely on binary outcome rewards when applying active sampling; for instance, PRIME (Cui et al., 2025) uses process rewards for RL finetuning but applies an accuracy-based sampling filter as in DS. Thus, a key open challenge in process-reward settings is to first establish a meaningful link between process rewards and sample informativeness, which would enable DPS or other sampling strategies to be applied in a principled way.

We conduct a preliminary investigation of applying DPS to continuous process rewards based on a simple hypothesis: prompts whose average trajectory returns fall into an intermediate range may be more informative. Specifically, we compute a return for each response by summing its process rewards, and then categorize each prompt's average return into one of three intervals defined by two boundaries, aiming to prioritize prompts in the middle interval. Using PRIME (Cui et al., 2025) as the testbed, we explore two DPS variants. The first uses fixed boundaries: since PRIME augments outcome rewards with small implicit process rewards, we simply set the boundaries to $0$ and $1$. The second uses dynamic, quantile-based boundaries, estimated from observed returns using quantiles $0.2$ and $0.8$, and updated via an exponential moving average (smoothing factor $0.9$). As shown in Figure 19, the dynamic-boundary variant outperforms both the fixed-boundary variant and US on Countdown and also increases the proportion of partially solved prompts in training batches. This improvement is likely due to the ability of dynamic boundaries to mitigate potential issues such as interval mismatches and sparse observations that may arise under fixed boundaries. We leave the development of more refined process-reward-based active sampling strategies for future work.

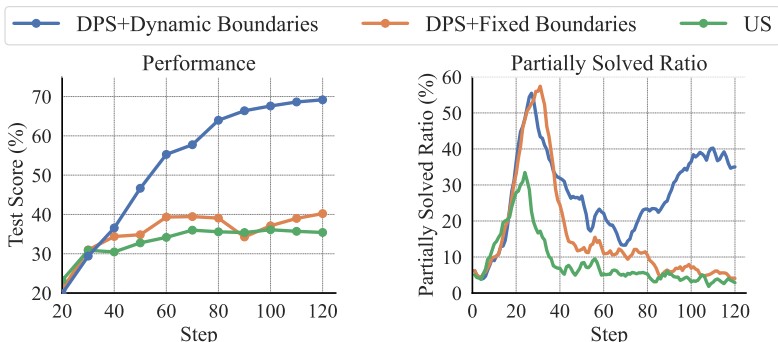

Figure 19: Evaluation in a precess reward setting on the Countdown 3B task. Sampling strategies are applied to the PRM-based method PRIME (response group $k = 4$, base RL algorithm RLOO).

## F DATA EXAMPLES

We provide below the illustrative data examples for each of the tasks in our experiments. Prompt templates for MATH and Geometry3k are drawn from verl (Sheng et al., 2024), whereas Countdown employs the template in Pan et al. (2025).

---

**MATH Data Example**

**Prompt:**
Given a prime $p$ and an integer $a$, we say that $a$ is a *primitive root* $\pmod p$ if the set $\{a, a^2, a^3, \ldots, a^{p-1}\}$ contains exactly one element congruent to each of $1, 2, 3, \ldots, p-1$ $\pmod p$.
For example, 2 is a primitive root $\pmod 5$ because $\{2, 2^2, 2^3, 2^4\} \equiv \{2, 4, 3, 1\} \pmod 5$, and this list contains every residue from 1 to 4 exactly once.
However, 4 is not a primitive root $\pmod 5$ because $\{4, 4^2, 4^3, 4^4\} \equiv \{4, 1, 4, 1\} \pmod 5$, and this list does not contain every residue from 1 to 4 exactly once.
What is the sum of all integers in the set $\{1, 2, 3, 4, 5, 6\}$ that are primitive roots $\pmod 7$?
Let's think step by step and output the final answer within \boxed{}.
**Answer:**
8

---

**Countdown Data Example**

**Prompt:**
A conversation between User and Assistant. The user asks a question, and the Assistant solves it. The assistant first thinks about the reasoning process in the mind and then provides the user with the answer.
User: Using the numbers [63, 95, 96], create an equation that equals 64. You can use basic arithmetic operations (+, -, *, /) and each number can only be used once. Show your work in <think> < /think> tags. And return the final answer in <answer> < /answer> tags, for example <answer> $(1 + 2)/3$ </answer>.
Assistant: Let me solve this step by step.
<think>

---

**Geometry3k Data Example**

**Prompt:**

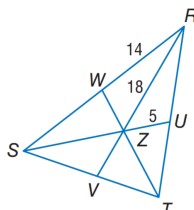

In $\triangle RST$, $Z$ is the centroid and $RZ = 18$. Find $ZV$. You FIRST think about the reasoning process as an internal monologue and then provide the final answer. The reasoning process MUST BE enclosed within <think> < /think> tags. The final answer MUST BE put in \boxed{}.
**Answer:**
9

---

# G    STATEMENT ON LLM USAGE

This work was completed without any substantive contribution of large language models (LLMs). The authors used LLMs exclusively for post-writing refinement. All core aspects of this work, including research ideation, methodology development, theoretical derivation, code implementation, experiments execution, and results analysis, were conceived and conducted solely by the authors.

