# OpenReview forum: "Dynamics-Predictive Sampling for Active RL Finetuning of Large Reasoning Models"
_ICLR.cc/2026/Conference — ICLR 2026 Poster_

### Official Review · Reviewer_joQK · 2025-10-24

**Soundness:** 3
**Presentation:** 3
**Contribution:** 3
**Rating:** 6
**Confidence:** 4

**Summary:**

This paper presents an online prompt selection method to improve the efficiency and the effectiveness of RLVR. The main motivation is to keep the informative prompts for training (those that yield both correct and wrong responses for the current policy). Instead of following previous work such as DAPO which may need sampling a big number of rollouts to obtain responses that have different rewards, this paper proposes a method named DPS to perform online Bayesian inference to estimate the state distributions. Three states represent, for a single prompt: all questions are incorrect, all are correct, and the rest. The prompts with the highest probabilities in the "partially solved" state will be selected for training.
For evaluation, the authors compared with several selection strategies, such as uniform sampling, history resampling (discard all prompts once the responses are all correct), and the "oracle" DS used in algorithms such as DAPO. The results on several reasoning benchmarks show the effectiveness and efficiency.

**Strengths:**

+ Compared to DS, this method achieves comparable performance yet with significantly fewer rollouts.
+ This paper addresses a key problem in RLVR about how to effectively select data.

**Weaknesses:**

+ In each training step, the state belief for each prompt in D is updated, which makes it less efficient than baseline sampling methods. This may be a problem when data scales, although the training data used in this work is usually relatively small-scale.
+ The authors did not show the method's sensitivity to the number of rollouts per prompt. It is unclear whether this method is still stable with a smaller number of rollouts.
+ The baselines are relatively naive and rule-based. More advanced methods, such as entropy/diversity/gradient-based selection, should be considered.
+ The training data and evaluation benchmarks are primarily math-related. The authors may consider general-domain tasks for evaluation.

**Questions:**

other suggestions:

+ The models are Qwen-series. The authors are suggested to test the performance on a model in other model families.
+ The authors may need to clarify that "oracle" actually refers to finding all prompts that have both successful and failed responses, instead of a method that can achieve the best performance by training on sampled prompts.

---

> ### Author Response · Authors · 2025-11-21
> **Rebuttal by Authors (Part 1/2)**
>
> We appreciate the time and effort you have dedicated to providing feedback on our paper and are grateful for the meaningful comments.
>
> **Q1: Computational efficiency when data scales.**
>
> Thanks for the thoughtful comment. We conduct additional experiments to examine how the computational cost of different operations scales with both dataset size and LLM size.
>
> (1) Computational scaling with dataset size. We construct pseudo-datasets (with arbitrary size $|\mathcal D|$) to more systematically evaluate the cost of DPS sampling and updates. Specifically, at each step $t$, we randomly generate $|\mathcal D|$ transition posterior matrices $\alpha_t^\tau$ and belief vectors $\mu_t^{\tau,\text{post}}$ and $\mu_t^{\tau,\text{prior}}$ corresponding to all $|\mathcal D|$ pseudo-samples. In the sampling stage, we perform top-$B$ selection on $\mu_t^{\tau,\text{prior}}(2)$; in the HMM-update stage, we assign random observations to the batch of $B$ samples and apply independent HMM updates to all $|\mathcal D|$ samples. For comparison, the per-step costs of LLM training and generation, which are independent of dataset size, are obtained by finetuning the 7B model on MATH. The new **Table 5** (page 23) reports the per-step costs of different operations for dataset sizes ranging from $10^4$ to $10^7$. The runtime and memory usage of DPS scale approximately linearly with dataset size, yet even for a very large dataset of size $|\mathcal D|=10^7$, DPS requires only 2.4s of runtime and 0.9 GiB of memory.  In contrast, LLM training and generation together require about 1100s of runtime and 600 GiB of GPU memory.
>
> Given its linear scaling, the computational overhead of DPS could become non-negligible at a sufficiently large scale ($|\mathcal D| > 10^8$), though such dataset sizes are beyond typical practical settings. For these cases, Appendix B (Time Complexity) also discusses a scheme that approximates the full-dataset updates and selection using a randomly sampled candidate subset $\hat {\mathcal B}$ satisfying $B < |\hat {\mathcal B}| \ll |\mathcal D|$.
>
> (2) Computational scaling with LLM size. The cost of LLM training and generation scales with model size. In particular, the additional rollout cost of DS also grows with LLM size, whereas DPS, as a rollout-free alternative to DS, incurs no such dependence. The new **Table 6** (page 23) compares the per-step costs of different operations for 1.5B and 7B models. The total runtime of LLM training and generation increases from roughly 370s to 1100s as the model size increases from 1.5B to 7B. At the 7B scale, the additional overhead introduced by DS versus DPS is approximately 1500s vs. 0.003s. Therefore, the advantage of DPS can become increasingly significant as LLM size grows.
>
> **Q2: Sensitivity analysis on the response group size k.**
>
> Thank you for this constructive comment.
>
> (1) Experiments. We conduct additional experiments using DPS and US under different response group sizes $k\in\\{4,8,16\\}$. The new **Figs. 13-14** (page 24) present the learning curves of test accuracy, effective sample ratio, and DPS prediction accuracy. The results show that DPS consistently outperforms US with both higher performance and effective sample ratios, and the advantage of DPS is the most pronounced when $k=4$. For US, the performance with $k=4$ drops substantially compared to $k=8$ and $16$ (falling to less than half), whereas DPS exhibits only a slight decrease (about $4\\%$). In particular, for $k=4$, the test accuracy of DPS is more than twice that of US.
>
> (2) Analysis of the significant advantage at small $k$. We attribute this to the fact that for smaller $k$, the probability that the same policy produces a mix of correct and incorrect responses for the same prompt becomes lower (given a fixed success rate $p$, the probability of generating mixed responses is $1 - p^k - (1-p)^k$). Hence, with a smaller $k$, the default US is much less likely to sample effective prompts (reflected in the extremely low effective sample ratio at $k = 4$ in **Fig. 14**). This creates greater potential for improvement when using DPS, which actively selects effective prompts. On the other hand, a smaller $k$ may lead to more frequent state transitions and make the underlying dynamics harder to estimate. Nevertheless, as shown in **Fig. 14**, DPS maintains high prediction accuracy at the small-yet-practical value of $k = 4$, leading to substantial performance gains. Consequently, in scenarios where the response group size is constrained, such as under limited training resources, applying DPS is likely to be particularly advantageous.
>
> **Due to the space constraints, please refer to the next block. Thanks!**

---

> ### Author Response · Authors · 2025-11-21
> **Rebuttal by Authors (Part 2/2)**
>
> ------Thank you for continuing to read!------
>
> **Q3: More advanced baselines, such as entropy/diversity/gradient-based selection, should be considered.**
>
> Thank you for the constructive comment. Following your suggestion, we implement a baseline that performs active sampling based on batch-level sample diversity. Specifically, we first pre-sample a candidate batch that is $n$ times larger than the actual training batch, and embed each prompt into a 1024-dimensional vector using WordLlama. We then iteratively select the candidate whose embedding maximizes the cumulative pairwise $L_2$ distance to previously selected samples, thereby greedily constructing a batch with high dispersion in the embedding space. We evaluate this variant on Countdown and tune the candidate batch size multiplier $n \in \\{2,4,8\\}$, ultimately selecting $n = 4$ as it yields slightly better performance. As shown in the new **Fig. 18** (page 26), diversity-based sampling offers only marginal improvements over US, and both its test accuracy and effective sample ratio remain far below those of DPS.
>
> Since our work approaches active sampling from the perspective of the connection between rewards and sample informativeness, DS corresponds to an "oracle" strategy aligned with this perspective; thus, our comparisons focus mainly on DS. DPS achieves comparable performance while requiring significantly fewer rollouts.
>
> **Q4: The authors may consider general-domain tasks for evaluation.**
>
> Thanks for the valuable suggestion. We additionally evaluate the MATH-trained models on general reasoning benchmarks, including ARC-c [1] and MMLU-Pro [2]. The results are provided in the new **Table 7** (page 27). On these general (OOD) reasoning tasks, DPS also shows consistent improvements over the baseline methods.
>
> **Q5: The authors are suggested to test the performance on a model in other model families.**
>
> Thanks for the valuable suggestion. We further train Llama-3.2-3B-Instruct on Countdown to evaluate different sampling methods. The new **Fig. 19** (page 28) compares the resulting test accuracies and effective sample ratios. The results show that, with Llama-3.2-3B-Instruct, DPS also performs comparably to DS and surpasses HR and US in both test accuracy and effective sample ratios, with even larger relative gains than those observed with the Qwen models.
>
> **Q6: Clarify the meaning of "oracle".**
>
> Thanks for the helpful comments. We have added a clarification of the meaning of "oracle" in the baseline description:
>
> > Line 308: Here, "oracle" refers to sampling a batch of all partially solved prompts, instead of achieving the best performance by training on sampled prompts.
>
> **Reference**
>
> [1] Clark et al. Think you have solved question answering? try arc, the ai2 reasoning challenge. 2018.
>
> [2] Wang et al. Mmlu-pro: A more robust and challenging multi-task language understanding benchmark. NeurIPS 2024.

---

### Official Review · Reviewer_xSRN · 2025-10-28

**Soundness:** 3
**Presentation:** 3
**Contribution:** 2
**Rating:** 6
**Confidence:** 2

**Summary:**

This paper proposes Dynamics-Predictive Sampling (DPS), a novel method to reduce the high computational cost of online prompt selection during the reinforcement learning (RL) finetuning of large reasoning models. The core problem is that existing methods, like Dynamic Sampling (DS), must perform extensive and costly LLM rollouts on large candidate batches to identify informative, "partially solved" training examples. DPS avoids this by modeling each prompt's solving progress as a dynamical system, specifically a hidden Markov model (HMM). This HMM tracks the prompt's latent "solving state" $z_{t}^{\tau}$ (e.g., fully unsolved, partially solved, or fully solved). Using historical reward signals, DPS performs online Bayesian inference to efficiently compute a predictive prior $\mu_{t}^{\tau,prior}(2)$, estimating the probability that a prompt is in the most informative "partially solved" state before any rollouts are generated. The main contributions are this HMM-based predictive framework and empirical results showing DPS substantially reduces redundant rollouts (e.g., achieving strong performance with less than 30% of DS's rollout budget), accelerates training, and achieves comparable or superior reasoning performance to costly oracle methods across diverse mathematics, planning, and visual geometry tasks.

**Strengths:**

The paper's primary strength is its direct and effective focus on improving the computational efficiency of active prompt selection for RL finetuning. The work is presented with good clarity, first identifying the high cost of existing online sampling methods (like DS) that rely on extensive rollouts, and then proposing a clear, lightweight alternative. In terms of originality, the paper offers a practical methodological refinement; rather than introducing a new sampling goal, it iteratively improves the mechanism by using an HMM to predict a prompt's solving state, thereby avoiding the need for costly pre-evaluation. The quality of this contribution is substantiated by a solid empirical evaluation across several reasoning tasks. The results consistently show that the proposed DPS method achieves performance comparable to the compute-intensive DS baseline while operating with a significantly reduced rollout budget, as detailed in the tables and efficiency graphs. The work's significance is, therefore, practical: it provides a viable method to achieve the benefits of dynamic data curation without the prohibitive computational overhead, making active sampling a more feasible option for practitioners.

**Weaknesses:**

* The necessity of the HMM framework is not fully justified, as the paper omits comparisons to simpler predictive baselines. A non-probabilistic heuristic, such as tracking an exponential moving average of reward variance for each prompt, might also identify "partially solved" items with similar efficiency gains and less modeling overhead.
* The claim of "negligible" computational overhead is only validated on relatively small datasets (e.g., 7.5k for MATH). The method's costs scale linearly with the total dataset size |D|, as it maintains and updates a separate HMM for every prompt, which could become a practical bottleneck for datasets with millions of examples.
* The Top-B selection strategy is purely exploitative, as it only samples prompts with the highest predicted probability of being in State 2. This greedy approach ignores sampling based on uncertainty (e.g., high entropy across the HMM state belief), which could be crucial for efficiently detecting state transitions and improving the model's robustness.

**Questions:**

Could you provide a justification for choosing an HMM framework over simpler, non-probabilistic heuristics? For instance, did you experiment with tracking a simple exponential moving average of reward variance for each prompt to identify "partially solved" items, and how would its performance and efficiency compare to DPS?

The Top-B selection strategy is greedy, exploiting prompts known to be in State 2. Did you consider alternative selection strategies that incorporate exploration, such as prioritizing prompts with high entropy in their predicted state distribution (i.e., high uncertainty), and how might that impact learning stability and final performance?

The definition of the solving states (unsolved, partial, solved) seems highly dependent on the number of samples, k=8. How does the model's performance and the HMM's state estimation change with different values of k (e.g., k=4 or k=16)?

You briefly mention that the framework could extend to dense or process-based rewards. Could you elaborate on how you would concretely map a continuous reward signal onto the discrete 3-state system? For example, would you use fixed thresholds, and how would you prevent this from re-introducing the observation-sparsity issues you noted in the ablation study on finer-grained partitions?

---

> ### Author Response · Authors · 2025-11-21
> **Rebuttal by Authors (Part 1/3)**
>
> We appreciate the time and effort you have dedicated to providing feedback on our paper and are grateful for the meaningful comments.
>
> **Q1: Comparisons to simpler predictive baselines, such as a non-probabilistic heuristic.**
>
> (1) Experiments. Thank you for the constructive comment. We implemented this variant, denoted Var+EMA, which tracks an exponential moving average of the reward variance for each prompt and samples the prompts with the top-B values across the dataset. We conduct experiments on Countdown with Var+EMA, tuning the EMA smoothing factor in $\\{0,0.1,0.5,0.9\\}$ and choosing $0.5$ as it yields relatively better performance. The comparative results in the new **Fig. 18** (page 26) show that DPS outperforms Var+EMA with consistently higher test accuracy and effective sample ratios. The following analyzes the necessity and advantages of the HMM framework over this simple predictive heuristic.
>
> (2) Analyses. (i) Dynamics estimation. Var+EMA implicitly assumes that the solving extent of each prompt tends to persist across steps, which resembles maintaining a fixed, stability-promoting transition model in DPS. Therefore, this heuristic is less flexible than DPS in capturing more complex underlying dynamics that may arise in practice. (ii) State prediction. Due to the infrequent sampling of a given prompt, its reward-variance observations are unavailable on most steps. Under this setting, Var+EMA lacks a reliable mechanism to extrapolate and predict variance during these unobserved intervals. In contrast, a core advantage of the HMM framework is its ability to model state transitions and, crucially, to extrapolate under missing observations, which can be particularly beneficial in batch-style RL finetuning.
>
> (3) Modeling overhead comparison. For each prompt, DPS maintains a $3\times 3$ matrix and two $3$-dimensional vectors, whereas Var+EMA maintains a single scalar. At each step, DPS performs $3$-dimensional matrix operations per prompt, while EMA performs scalar updates. In response to **Q2**, we additionally report experiments on how the computational cost of DPS scales with dataset size. Regarding hyperparameters, DPS uses only one parameter, the non-stationary decay parameter $\lambda$, whereas Var+EMA uses an EMA smoothing factor.
>
> **Q2: Computational efficiency when data scales.**
>
> Thanks for the thoughtful comment. We conduct additional experiments to examine how the computational cost of different operations scales with both dataset size and LLM size.
>
> (1) Computational scaling with dataset size. We construct pseudo-datasets (with arbitrary size $|\mathcal D|$) to more systematically evaluate the cost of DPS sampling and updates. Specifically, at each step $t$, we randomly generate $|\mathcal D|$ transition posterior matrices $\alpha_t^\tau$ and belief vectors $\mu_t^{\tau,\text{post}}$ and $\mu_t^{\tau,\text{prior}}$ corresponding to all $|\mathcal D|$ pseudo-samples. In the sampling stage, we perform top-$B$ selection on $\mu_t^{\tau,\text{prior}}(2)$; in the HMM-update stage, we assign random observations to the batch of $B$ samples and apply independent HMM updates to all $|\mathcal D|$ samples. For comparison, the per-step costs of LLM training and generation, which are independent of dataset size, are obtained by finetuning the 7B model on MATH. The new **Table 5** (page 23) reports the per-step costs of different operations for dataset sizes ranging from $10^4$ to $10^7$. The runtime and memory usage of DPS scale approximately linearly with dataset size, yet even for a very large dataset of size $|\mathcal D|=10^7$, DPS requires only 2.4s of runtime and 0.9 GiB of memory.  In contrast, LLM training and generation together require about 1100s of runtime and 600 GiB of GPU memory.
>
> Given its linear scaling, the computational overhead of DPS could become non-negligible at a sufficiently large scale ($|\mathcal D|>10^8$), though such dataset sizes are beyond typical practical settings. For these cases, Appendix B (Time Complexity) also discusses a scheme that approximates the full-dataset updates and selection using a randomly sampled candidate subset $\hat{\mathcal B}$ satisfying $B<|\hat {\mathcal B}|\ll|\mathcal D|$.
>
> (2) Computational scaling with LLM size. The cost of LLM training and generation scales with model size. In particular, the additional rollout cost of DS also grows with LLM size, whereas DPS, as a rollout-free alternative to DS, incurs no such dependence. The new **Table 6** (page 23) compares the per-step costs of different operations for 1.5B and 7B models. The total runtime of LLM training and generation increases from roughly 370s to 1100s as the model size increases from 1.5B to 7B. At the 7B scale, the additional overhead introduced by DS versus DPS is approximately 1500s vs. 0.003s. Therefore, the advantage of DPS can become increasingly significant as LLM size grows.
>
> **Due to the space constraints, please refer to the next block. Thanks!**

---

> ### Author Response · Authors · 2025-11-21
> **Rebuttal by Authors (Part 2/3)**
>
> ------Thank you for continuing to read!------
>
> **Q3:  The Top-B selection strategy is purely exploitative.**
>
> (1) Experiments. Thanks for this insightful comment. We agree that introducing exploration into sample selection could potentially improve the model's robustness. Prior to settling on the final DPS design, we actually tested a variant called DPS+Entropy, which explicitly balances exploitation and exploration by combining the entropy of the predicted distribution with the State-2 probability for Top-B sampling. We conducted experiments on Countdown and tuned the entropy regularization coefficient in $\\{0.01,0.1,1,10\\}$. The training curves are shown in the new **Fig. 15** (page 25). DPS+Entropy performs best when the coefficient is $0.1$, but it does not yield a noticeably stronger improvement over DPS in either test accuracy or effective sample ratio. We provide further analysis below.
>
> (2) Analysis. While the Top-B selection strategy is purely exploitative, it exploits an objective (i.e., the predicted probability) that already incorporates a degree of exploration. The non-stationary decay mechanism in DPS (Eq. (15)), although originally designed to accommodate non-stationary dynamics, also implicitly introduces exploration. It gradually decays the transition posterior and drifts the predicted states of under-sampled prompts (i.e., those predicted to be in State 1 or 3) toward a more uniform distribution, increasing their likelihood of being selected and updated. This behavior is supported by the new **Fig. 16** (page 25), which shows that a smaller decay parameter $\lambda$ leads to more uniform sample counts, with a lower variance and a higher minimum across the dataset. Hence, the additional entropy term partially overlaps with this built-in exploration effect, which may account for the limited improvement.
>
> (3) Finally, the core contribution of our method lies in modeling and predicting the dynamics of solving states. Once the state distribution is predicted, any selection criterion, such as softmax selection or entropy-based sampling, can be applied. Therefore, the specific choice of selection criterion is not the primary focus of this work. DPS adopts a simple criterion and introduces as few hyperparameters as possible (with only $\lambda$) while already achieving strong performance.
>
> **Q4: How does the model's performance and the HMM's state estimation change with different values of k (e.g., k=4 or k=16)?**
>
> Thank you for this valuable question.
>
> (1) Experiments. We conduct additional experiments using DPS and US under different response group sizes $k\in\\{4,8,16\\}$. The new **Figs. 13-14** (page 24) present the learning curves of test accuracy, effective sample ratio, and DPS prediction accuracy. The results show that DPS consistently outperforms US with both higher performance and effective sample ratios, and the advantage of DPS is the most pronounced when $k=4$. For US, the performance with $k=4$ drops substantially compared to $k=8$ and $16$ (falling to less than half), whereas DPS exhibits only a slight decrease (about $4\\%$). In particular, for $k=4$, the test accuracy of DPS is more than twice that of US.
>
> (2) Analysis of the significant advantage at small $k$. We attribute this to the fact that for smaller $k$, the probability that the same policy produces a mix of correct and incorrect responses for the same prompt becomes lower (given a fixed success rate $p$, the probability of generating mixed responses is $1 - p^k - (1-p)^k$). Hence, with a smaller $k$, the default US is much less likely to sample effective prompts (reflected in the extremely low effective sample ratio at $k = 4$ in **Fig. 14**). This creates greater potential for improvement when using DPS, which actively selects effective prompts. On the other hand, a smaller $k$ may lead to more frequent state transitions and make the underlying dynamics harder to estimate. Nevertheless, as shown in **Fig. 14**, DPS maintains high prediction accuracy at the small-yet-practical value of $k = 4$, leading to substantial performance gains. Consequently, in scenarios where the response group size is constrained, such as under limited training resources, applying DPS is likely to be particularly advantageous.

---

> ### Author Response · Authors · 2025-11-21
> **Rebuttal by Authors (Part 3/3)**
>
> ------Thank you for continuing to read!------
>
> **Q5: Extension to continuous process rewards.**
>
> Thank you for the thoughtful question. A key challenge for applying principled active sampling in process-reward settings is to first establish a meaningful link between process rewards and sample informativeness, which remains an open problem. We conduct a preliminary investigation of applying DPS to continuous process rewards based on a simple hypothesis: prompts whose average trajectory returns fall into an intermediate range may be more informative. Specifically, we compute a return for each response by summing its process rewards, and then categorize each prompt's average return into one of three intervals defined by two boundaries, aiming to prioritize prompts in the middle interval. Using PRIME [1] as the testbed, we explore two DPS variants. The first uses fixed boundaries: since PRIME augments outcome rewards with small implicit process rewards, we simply set the boundaries to $0$ and $1$. The second uses dynamic, quantile-based boundaries, estimated from observed returns using quantiles $0.2$ and $0.8$, and updated via an exponential moving average (smoothing factor $0.9$). As shown in the new **Fig. 20** (page 28), the dynamic-boundary variant outperforms both the fixed-boundary variant and uniform sampling (US) on Countdown and also increases the proportion of partially solved prompts in training batches. This improvement is likely due to the ability of dynamic boundaries to mitigate potential issues such as interval mismatches and sparse observations that may arise under fixed boundaries. We leave the development of more refined process-reward-based active sampling strategies for future work.
>
> **Reference**
>
> [1] Cui et al. Process reinforcement through implicit rewards. 2025.

---

> > ### Comment · Reviewer_xSRN · 2025-11-22
> >
> > I'd like to thank the authors for engaging with my concerns and extensive rebuttal responses. Now I see more clearly the benefits of the proposed Dynamics-Predictive Sampling.

---

> > > ### Author Response · Authors · 2025-11-23
> > >
> > > Thank you for the positive feedback. We appreciate your thoughtful comments and are glad that our response has made the benefits of DPS clearer. If there are any remaining points that might benefit from further clarification, we would be very happy to elaborate.

---

### Official Review · Reviewer_Nhzv · 2025-10-29

**Soundness:** 2
**Presentation:** 3
**Contribution:** 2
**Rating:** 4
**Confidence:** 3

**Summary:**

This paper introduces Dynamics-Predictive Sampling (DPS), a method that predicts the informativeness of training samples by modeling their learning progression as a Hidden Markov Model (HMM), thereby replacing costly LLM rollouts for sample selection. The method's primary strength is its computational efficiency, shifting the overhead from LLM inference to lightweight matrix operations. This claim is well-supported by comprehensive experiments across diverse reasoning tasks.
However, its limitations warrant attention. First, its three-state model is tailored for binary rewards, and its generality for tasks with denser reward signals (e.g., from process supervision) is unclear. Second, by maintaining independent dynamics for each prompt, the method may suffer from data sparsity issues, leading to unreliable estimates for infrequently sampled prompts and thus impacting its robustness.

**Strengths:**

- The core contribution lies in shifting the sample selection overhead from LLM inference costs, which are proportional to batch size, to computationally negligible matrix operations. Experiments convincingly show that DPS can match or exceed the performance of baselines with substantially lower computational resources, a critical factor for the practical adoption of RL fine-tuning.
- The study features a comprehensive experimental design across diverse and complex reasoning tasks (mathematics, planning, visual geometry), with thorough comparisons against key baselines (e.g., Uniform Sampling, Dynamic Sampling). The inclusion of extensive ablation studies enhances the credibility of the methodological design and provides good support for the core arguments.

**Weaknesses:**

- The proposed three-state model (unsolved, partially solved, solved) is highly effective for binary reward settings. It would be beneficial to discuss the framework's extensibility to tasks with denser reward signals, such as continuous scores from process supervision. Addressing questions like how the state space could be partitioned (e.g., fixed vs. dynamic boundaries) would provide valuable insight into the method's potential applicability to a wider range of problems.
- The model maintains independent transition dynamics for each prompt. This raises a concern about data sparsity: for infrequently sampled prompts, the estimated dynamics may be unreliable, which could compromise the accuracy of state predictions. It would strengthen the paper to explain how this potential issue is handled. Additionally, an analysis of what happens to persistently under-sampled, difficult problems—and whether their learning dynamics risk stagnation—would be a valuable addition.

**Questions:**

- The state observation depends on generating k responses per prompt, creating a trade-off between cost and observation noise. The paper uses k=8 but does not include a sensitivity analysis for this hyperparameter. An analysis or discussion on the impact of varying k—and how effectively the HMM can mitigate noise from smaller k values—would provide a clearer picture of the method's robustness.

---

> ### Author Response · Authors · 2025-11-21
> **Rebuttal by Authors (Part 1/2)**
>
> We appreciate the time and effort you have dedicated to providing feedback on our paper and are grateful for the meaningful comments.
>
> **Q1: Extension to continuous process rewards.**
>
> Thank you for this constructive comment. We agree that extending DPS to more general reward signals is a valuable direction. Below, we first discuss the main challenges of applying active sampling in process-reward settings, and then present our preliminary exploration of extending DPS to continuous process rewards.
>
> (1) Our focus on binary rewards reflects their practical prevalence and their well-understood connection with sample informativeness, which enables principled and interpretable sampling strategies. In contrast, how process rewards relate to informativeness remains unclear in the field. To our knowledge, existing methods that incorporate process rewards still rely on binary outcome rewards when applying active sampling; for instance, PRIME [1] uses process rewards for RL finetuning but applies an accuracy-based sampling filter as in DS. Thus, a key open challenge in process-reward settings is to first establish a meaningful link between process rewards and sample informativeness, which would enable DPS or other sampling strategies to be applied in a principled way.
>
> (2) We conduct a preliminary investigation of applying DPS to continuous process rewards based on a simple hypothesis: prompts whose average trajectory returns fall into an intermediate range may be more informative. Specifically, we compute a return for each response by summing its process rewards, and then categorize each prompt's average return into one of three intervals defined by two boundaries, aiming to prioritize prompts in the middle interval. Using PRIME [1] as the testbed, we explore two DPS variants. The first uses fixed boundaries: since PRIME augments outcome rewards with small implicit process rewards, we simply set the boundaries to $0$ and $1$. The second uses dynamic, quantile-based boundaries, estimated from observed returns using quantiles $0.2$ and $0.8$, and updated via an exponential moving average (smoothing factor $0.9$). As shown in the new **Fig. 20** (page 28), the dynamic-boundary variant outperforms both the fixed-boundary variant and uniform sampling (US) on Countdown and also increases the proportion of partially solved prompts in training batches. This improvement is likely due to the ability of dynamic boundaries to mitigate potential issues such as interval mismatches and sparse observations that may arise under fixed boundaries. We leave the development of more refined process-reward-based active sampling strategies for future work.
>
> **Due to the space constraints, please refer to the next block. Thanks!**

---

> ### Author Response · Authors · 2025-11-21
> **Rebuttal by Authors (Part 2/2)**
>
> ------Thank you for continuing to read!------
>
> **Q2: About data sparsity and prediction accuracy on certain prompts.**
>
> Thanks for the thoughtful comment and suggestions.
>
> (1) How data sparsity is handled in DPS. We agree that infrequently sampled prompts can have relatively inaccurate state predictions, which, without intervention, may reduce their chance of being sampled and create a negative feedback loop. However, the non-stationary decay mechanism of DPS (Eq. (15)), although originally designed to accommodate potentially non-stationary dynamics, effectively introduces an exploratory behavior that mitigates this risk. By gradually decaying the transition posterior, the state predictions of under-sampled prompts (typically those in State 1 or 3) drift toward a uniform distribution, increasing the likelihood that they will be resampled. When the model can no longer identify informative prompts with clearly higher probabilities of being partially solved, these under-sampled prompts naturally get selected again, allowing their state predictions to be updated. This behavior is supported by the new **Fig. 16** on page 25, which shows that a smaller decay parameter $\lambda$ leads to more uniform sample counts, with a lower variance and a higher minimum across the dataset.
>
> (2) Prediction accuracies for specific types of prompts. To further examine how DPS estimates states for persistently under-sampled or difficult prompts, we conducted the following diagnostic evaluation after each training step. We sampled: (a) the 256 prompts with the fewest past sample counts (under-sampled prompts), (b) the 256 prompts with the highest DPS-estimated probability of being fully unsolved (most likely persistently difficult prompts), and (c) 256 randomly selected prompts. We then rolled out these prompts without training the policy and without updating the HMM, purely to measure prediction accuracy. The new **Fig. 17** (page 26) shows their prediction accuracies on Countdown. We observe that difficult prompts achieve even higher prediction accuracy than uniform prompts, likely because the difficult ones often have simpler or more stable state distributions and transitions, making them easier to predict even with fewer observations. Under-sampled prompts typically show slightly lower accuracy than uniform prompts, but the gap is small. We hypothesize that the same reason applies here: under-sampled prompts in DPS correspond to those confidently predicted to be in State 1 or 3, and thus may consist largely of very hard or very easy problems, which are generally easier to predict. In this sense, the prompts most susceptible to estimation error under infrequent sampling are, under DPS's mechanism, often those whose states are inherently easier to infer. This provides an additional perspective on why DPS could mitigate the impact of data sparsity.
>
> **Q3: Sensitivity analysis on the response group size k.**
>
> Thank you for this constructive suggestion.
>
> (1) Experiments. We conduct additional experiments using DPS and US under different response group sizes $k\in\\{4,8,16\\}$. The new **Figs. 13-14** (page 24) present the learning curves of test accuracy, effective sample ratio, and DPS prediction accuracy. The results show that DPS consistently outperforms US with both higher performance and effective sample ratios, and the advantage of DPS is the most pronounced when $k=4$. For US, the performance with $k=4$ drops substantially compared to $k=8$ and $16$ (falling to less than half), whereas DPS exhibits only a slight decrease (about $4\\%$). In particular, for $k=4$, the test accuracy of DPS is more than twice that of US.
>
> (2) Analysis of the significant advantage at small $k$. We attribute this to the fact that for smaller $k$, the probability that the same policy produces a mix of correct and incorrect responses for the same prompt becomes lower (given a fixed success rate $p$, the probability of generating mixed responses is $1 - p^k - (1-p)^k$). Hence, with a smaller $k$, the default US is much less likely to sample effective prompts (reflected in the extremely low effective sample ratio at $k = 4$ in **Fig. 14**). This creates greater potential for improvement when using DPS, which actively selects effective prompts. On the other hand, a smaller $k$ may lead to more frequent state transitions and make the underlying dynamics harder to estimate. Nevertheless, as shown in **Fig. 14**, DPS maintains high prediction accuracy at the small-yet-practical value of $k = 4$, leading to substantial performance gains. Consequently, in scenarios where the response group size is constrained, such as under limited training resources, applying DPS is likely to be particularly advantageous.
>
> **Reference**
>
> [1] Cui et al. Process reinforcement through implicit rewards. 2025.

---

> > ### Comment · Reviewer_Nhzv · 2025-11-24
> > **Official Comment by Reviewer Nhzv**
> >
> > I thank the authors for their comprehensive rebuttal. The new experiments and analyses have effectively addressed my concerns and strengthened the manuscript. I have updated my evaluation accordingly.

---

> > > ### Author Response · Authors · 2025-11-24
> > >
> > > Thank you for the positive feedback and the update. We're glad to hear that our response has effectively addressed your concerns. Your comments and suggestions have been very helpful in strengthening the manuscript.

---

### Official Review · Reviewer_4vBK · 2025-10-31

**Soundness:** 3
**Presentation:** 3
**Contribution:** 2
**Rating:** 6
**Confidence:** 2

**Summary:**

The paper introduces Dynamics-Predictive Sampling (DPS), an online prompt-selection framework that enhances reinforcement-learning (RL) finetuning of large reasoning models (LRMs). The core motivation is that dynamic sampling strategies (e.g., Dynamic Sampling (DS), History Resampling (HR)) improve RL finetuning efficiency by prioritizing informative prompts—typically those that are only partially solved—but at the cost of significant rollout overhead.
DPS aims to predict which prompts are likely to be informative before performing expensive LLM rollouts. It models each prompt’s solving progress as a Hidden Markov Model (HMM) whose latent states correspond to unsolved, partially solved, and fully solved stages. Using Bayesian updates of transition and emission probabilities, DPS infers prompt-specific state distributions online and selects prompts with the highest predicted probability of being partially solved.

The proposed method is lightweight and scalable, requiring only low-dimensional updates per prompt. Experiments on mathematical reasoning (MATH, AIME24, AMC23), numerical planning (Countdown), and visual geometry (Geometry3K) demonstrate that DPS.

**Strengths:**

The paper offers a new dynamical-systems perspective on data selection in RL finetuning, bridging prompt-level reward evolution with state-space modeling.

DPS eliminates the need for redundant LLM rollouts by predicting informative prompts, yielding 2–3× runtime reduction and 70 % rollout savings while maintaining accuracy parity with DAPO.

Comprehensive experiments across three reasoning domains and multiple model scales. Evaluation includes accuracy curves, confusion-matrix analyses of prediction reliability, and ablation studies (non-stationary decay λ, number of states, prior α₀) demonstrating robustness and interpretability.

**Weaknesses:**

Reward formulation dependency:

The approach currently depends on binary correctness rewards, which are straightforward in math-style benchmarks but not directly transferable to open-ended or process-based domains (e.g., code synthesis with partial correctness). The authors acknowledge this and suggest extending DPS to dense or step-wise rewards.

Simplistic selection criterion:

DPS uses top-B selection on predicted State 2 probabilities. More nuanced criteria (e.g., entropy-based uncertainty, expected information gain) could further enhance exploration-exploitation balance.

Scalability in very large datasets:

While the complexity analysis asserts negligible overhead, maintaining per-prompt HMM parameters might become costly when |D| ≫ 10⁶ unless sampling approximations are used.

**Questions:**

None

---

> ### Author Response · Authors · 2025-11-21
> **Rebuttal by Authors (Part 1/2)**
>
> We appreciate the time and effort you have dedicated to providing feedback on our paper and are grateful for the meaningful comments.
>
> **Q1: Extension to continuous process rewards.**
>
> Thank you for the valuable comment. We agree that extending DPS to more general reward signals is a valuable direction. Below, we first discuss the main challenges of applying active sampling in process-reward settings, and then present our preliminary exploration of extending DPS to continuous process rewards.
>
> (1) Our focus on binary rewards reflects their practical prevalence and their well-understood connection with sample informativeness, which enables principled and interpretable sampling strategies. In contrast, how process rewards relate to informativeness remains unclear in the field. To our knowledge, existing methods that incorporate process rewards still rely on binary outcome rewards when applying active sampling; for instance, PRIME [1] uses process rewards for RL finetuning but applies an accuracy-based sampling filter as in DS. Thus, a key open challenge in process-reward settings is to first establish a meaningful link between process rewards and sample informativeness, which would enable DPS or other sampling strategies to be applied in a principled way.
>
> (2) We conduct a preliminary investigation of applying DPS to continuous process rewards based on a simple hypothesis: prompts whose average trajectory returns fall into an intermediate range may be more informative. Specifically, we compute a return for each response by summing its process rewards, and then categorize each prompt's average return into one of three intervals defined by two boundaries, aiming to prioritize prompts in the middle interval. Using PRIME [1] as the testbed, we explore two DPS variants. The first uses fixed boundaries: since PRIME augments outcome rewards with small implicit process rewards, we simply set the boundaries to $0$ and $1$. The second uses dynamic, quantile-based boundaries, estimated from observed returns using quantiles $0.2$ and $0.8$, and updated via an exponential moving average (smoothing factor $0.9$). As shown in the new **Fig. 20** (page 28), the dynamic-boundary variant outperforms both the fixed-boundary variant and uniform sampling (US) on Countdown and also increases the proportion of partially solved prompts in training batches. This improvement is likely due to the ability of dynamic boundaries to mitigate potential issues such as interval mismatches and sparse observations that may arise under fixed boundaries. We leave the development of more refined process-reward-based active sampling strategies for future work.
>
> **Q2: More nuanced selection criteria beyond top-B.**
>
> (1) Experiments. Thanks for this insightful suggestion. We agree that introducing exploration into sample selection could potentially improve the model's robustness. Prior to settling on the final DPS design, we actually tested a variant called DPS+Entropy, which explicitly balances exploitation and exploration by combining the entropy of the predicted distribution with the State-2 probability for Top-B sampling. We conducted experiments on Countdown and tuned the entropy regularization coefficient in $\\{0.01,0.1,1,10\\}$. The training curves are shown in the new **Fig. 15** on page 25. DPS+Entropy performs best when the coefficient is $0.1$, but it does not yield a noticeably stronger improvement over DPS in either test accuracy or effective sample ratio. We provide further analysis below.
>
> (2) Analysis. While the Top-B selection strategy is purely exploitative, it exploits an objective (i.e., the predicted probability) that already incorporates a degree of exploration. The non-stationary decay mechanism in DPS (Eq. (15)), although originally designed to accommodate non-stationary dynamics, also implicitly introduces exploration. It gradually decays the transition posterior and drifts the predicted states of under-sampled prompts (i.e., those predicted to be in State 1 or 3) toward a more uniform distribution, increasing their likelihood of being selected and updated. This behavior is supported by the new **Fig. 16** (page 25), which shows that a smaller decay parameter $\lambda$ leads to more uniform sample counts, with a lower variance and a higher minimum across the dataset. Hence, the additional entropy term partially overlaps with this built-in exploration effect, which may account for the limited improvement.
>
> (3) Finally, the core contribution of our method lies in modeling and predicting the dynamics of solving states. Once the state distribution is predicted, any selection criterion, such as softmax selection or entropy-based sampling, can be applied. Therefore, the specific choice of selection criterion is not the primary focus of this work. DPS adopts a simple criterion and introduces as few hyperparameters as possible (with only $\lambda$) while already achieving strong performance.

---

> ### Author Response · Authors · 2025-11-21
> **Rebuttal by Authors (Part 2/2)**
>
> ------Thank you for continuing to read!------
>
> **Q3: Scalability in very large datasets.**
>
> Thanks for the thoughtful comment. We conduct additional experiments to examine how the computational cost of different operations scales with both dataset size and LLM size.
>
> (1) Computational scaling with dataset size. We construct pseudo-datasets (with arbitrary size $|\mathcal D|$) to more systematically evaluate the cost of DPS sampling and updates. Specifically, at each step $t$, we randomly generate $|\mathcal D|$ transition posterior matrices $\alpha_t^\tau$ and belief vectors $\mu_t^{\tau,\text{post}}$ and $\mu_t^{\tau,\text{prior}}$ corresponding to all $|\mathcal D|$ pseudo-samples. In the sampling stage, we perform top-$B$ selection on $\mu_t^{\tau,\text{prior}}(2)$; in the HMM-update stage, we assign random observations to the batch of $B$ samples and apply independent HMM updates to all $|\mathcal D|$ samples. For comparison, the per-step costs of LLM training and generation, which are independent of dataset size, are obtained by finetuning the 7B model on MATH. The new **Table 5** (page 23) reports the per-step costs of different operations for dataset sizes ranging from $10^4$ to $10^7$. The runtime and memory usage of DPS scale approximately linearly with dataset size, yet even for a very large dataset of size $|\mathcal D|=10^7$, DPS requires only 2.4s of runtime and 0.9 GiB of memory.  In contrast, LLM training and generation together require about 1100s of runtime and 600 GiB of GPU memory.
>
> Given its linear scaling, the computational overhead of DPS could become non-negligible at a sufficiently large scale ($|\mathcal D| > 10^8$), though such dataset sizes are beyond typical practical settings. For these cases, Appendix B (Time Complexity) also discusses a scheme that approximates the full-dataset updates and selection using a randomly sampled candidate subset $\hat {\mathcal B}$ satisfying $B < |\hat {\mathcal B}| \ll |\mathcal D|$.
>
> (2) Computational scaling with LLM size. The cost of LLM training and generation scales with model size. In particular, the additional rollout cost of DS also grows with LLM size, whereas DPS, as a rollout-free alternative to DS, incurs no such dependence. The new **Table 6** (page 23) compares the per-step costs of different operations for 1.5B and 7B models. The total runtime of LLM training and generation increases from roughly 370s to 1100s as the model size increases from 1.5B to 7B. At the 7B scale, the additional overhead introduced by DS versus DPS is approximately 1500s vs. 0.003s. Therefore, the advantage of DPS can become increasingly significant as LLM size grows.
>
> **Reference**
>
> [1] Cui et al. Process reinforcement through implicit rewards. 2025.

---

### Author Response · Authors · 2025-11-22
**Global Response by Authors**

### **Global Response**

We thank all the reviewers for taking the time to read our manuscript carefully and for providing constructive and insightful feedback. We are encouraged by the positive comments, such as:

- Offering a new dynamical-systems perspective on data selection in RL finetuning (Reviewer 4vBK).
- Addressing a key problem in RLVR (Reviewer joQK) and demonstrating practical significance (Reviewers xSRN/Nhzv).
- Comparable or superior performance with substantially lower computational overhead (Reviewers 4vBK/Nhzv/xSRN/joQK).
- Comprehensive and solid evaluation (Reviewers 4vBK/Nhzv/xSRN) with extensive ablations (Reviewer Nhzv), showing robustness and interpretability (Reviewer 4vBK).
- Presented with good clarity (Reviewer xSRN).

Meanwhile, we have worked carefully to address the reviewers' concerns and questions, and we provide detailed responses to the individual reviews below. We have also updated the manuscript and included all additional experiments on pages 23-28. These include:

- Computational scaling behavior (Tables 5-6).
- Sensitivity to response group size and to additional entropy-regularized selection (Figs. 13-15).
- Extension to continuous process rewards (Fig. 20).
- Sampling behavior and prediction accuracy on specific types of prompts (Figs. 16-17).
- Additional baselines, benchmarks, and an extra model (Figs. 18-19, Table 7).

We hope our responses adequately address the reviewers' concerns. We would be very happy to clarify any remaining questions and look forward to further discussion.

---

### Author Response · Authors · 2025-12-03
**A Summary of Rebuttal and Discussions**

### A Summary of Rebuttal and Discussions

We sincerely thank the Area Chairs and Reviewers for their time, effort, and thoughtful feedback on our manuscript. Below, we summarize the shared concerns, our corresponding responses, and the post-rebuttal feedback to assist evaluation.

**Post-rebuttal comments and updates.**

During the discussion phase, Reviewer NHzv and Reviewer xSRN commented that our rebuttal "effectively addressed the concerns and strengthened the manuscript" and made "the benefits of DPS clearer", respectively. Reviewer NHzv also raised the score from **4 to 6** before the reviews were reverted. These updates occurred several days prior to the announcement of the system issue.

**Shared concerns/questions and summarized responses.**

Q1. Scalability in very large datasets (Reviewers xSRN, 4vBK, joQK)

We conduct systematic scaling experiments using pseudo-datasets up to size $10^7$ (new **Table 5,** page 23). DPS scales linearly and remains lightweight even at $10^7$ prompts (2.4s runtime, 0.9 GiB memory), far below LLM training and generation costs (∼1100s and 600 GiB). For even larger scales beyond typical practice, Appendix B also discusses an approximation scheme using randomly sampled candidate subsets per step.

We additionally analyze scaling with LLM size (new **Table 6,** page 23). The rollout cost of DS increases rapidly with model size, while DPS remains rollout-free. At 7B scale, DS vs. DPS adds ~1500s vs. 0.003s of extra overhead per step. Thus, the advantage of DPS can become increasingly significant as LLM size grows.

Q2. Extension to continuous process rewards (Reviewers Nhzv, 4vBK)

Our focus on binary rewards reflects their practical prevalence and their well-understood connection with sample informativeness, a link that is not yet clear for process rewards. Existing methods using process rewards (e.g., PRIME) still rely on binary outcome signals for active sampling. To explore this direction, we provide a preliminary extension of DPS to continuous process rewards, implementing both fixed and dynamic boundary variants based on PRIME. As shown in **Fig. 20** (page 28), the dynamic-boundary variant outperforms uniform sampling and fixed boundaries, suggesting that extending DPS is feasible.

Q3. Selection criteria beyond Top-B (Reviewers xSRN, 4vBK)

We implemented DPS+Entropy, which explicitly balances exploitation and exploration, and conducted experiments under different entropy coefficients. As shown in **Fig. 15** (page 25), this variant does not noticeably outperform DPS. This is because DPS already incorporates implicit exploration via its non-stationary decay mechanism (Eq. 15), which gradually increases sampling probability of under-sampled prompts, a behavior supported by **Fig. 16** (page 25). We note that DPS's core contribution lies in modeling and predicting solving-state dynamics; once predicted, any selection rule can be applied. DPS adopts a simple Top-B scheme to keep hyperparameters minimal.

Q4. Sensitivity analysis on the response group size k (Reviewer Nhzv, xSRN, joQK)

We conduct experiments for $k\in\\{4,8,16\\}$ under DPS and uniform sampling (new **Figs. 13-14**, page 24). DPS consistently outperforms US across all $k$, with the largest gains at the small-yet-practical $k=4$. Mathematically, a smaller k reduces the likelihood of generating mixed responses under a fixed success rate, greatly lowering the effective sample ratio of US (**Fig. 14**), while DPS maintains high prediction accuracy to select informative prompts. This suggests DPS may be particularly beneficial in resource-limited regimes where only small response groups can be generated.

Overall, this work proposes Dynamics-Predictive Sampling (DPS) for RL finetuning, which online predicts and selects informative prompts by inferring their learning dynamics. It eliminates the need for rollout-intensive filtering and substantially accelerates RL finetuning across diverse reasoning tasks.

We hope this summary is helpful, and we thank you again for your time and consideration.

---

### Meta-Review · Area_Chair_fP2q · 2026-01-07

**Summary:**

This paper proposes a novel reinforcement learning algorithm for prioritizing partially solved examples, which uses a dynamical system model to keep track of which examples have been solved so far.

**Reviewer Concerns:**

The reviewers largely found the work to be a strong contribution, but had concerns about scalability of the proposed technique as well as its generality. The authors provided additional experiments and discussion to address these concerns.

**Reviewer Scores:**

The negative reviewer raised their score, and at the end all the reviewers were positive about the paper.

---

### Decision · Program_Chairs · 2026-01-26

Accept (Poster)